# Targeting key angiogenic pathways with a bispecific CrossMAb optimized for neovascular eye diseases

Jörg T Regula[1,†], Peter Lundh von Leithner[2,†], Richard Foxton[2,3], Veluchamy A Barathi[4,5], Chui Ming Gemmy Cheung[4], Sai Bo Bo Tun[4], Yeo Sia Wey[4], Daiju Iwata[2], Miroslav Dostalek[3], Jörg Moelleken[1], Kay G Stubenrauch[1], Everson Nogoceke[3], Gabriella Widmer[3], Pamela Strassburger[3], Michael J Koss[6,7], Christian Klein[8], David T Shima[2] & Guido Hartmann[3,*]

## Abstract

Anti-angiogenic therapies using biological molecules that neutralize vascular endothelial growth factor-A (VEGF-A) have revolutionized treatment of retinal vascular diseases including age-related macular degeneration (AMD). This study reports preclinical assessment of a strategy to enhance anti-VEGF-A monotherapy efficacy by targeting both VEGF-A and angiopoietin-2 (ANG-2), a factor strongly upregulated in vitreous fluids of patients with retinal vascular disease and exerting some of its activities in concert with VEGF-A. Simultaneous VEGF-A and ANG-2 inhibition was found to reduce vessel lesion number, permeability, retinal edema, and neuron loss more effectively than either agent alone in a spontaneous choroidal neovascularization (CNV) model. We describe the generation of a bispecific domain-exchanged (crossed) monoclonal antibody (CrossMAb; RG7716) capable of binding, neutralizing, and depleting VEGF-A and ANG-2. RG7716 showed greater efficacy than anti-VEGF-A alone in a non-human primate laser-induced CNV model after intravitreal delivery. Modification of RG7716's FcRn and FcγR binding sites disabled the antibodies' Fc-mediated effector functions. This resulted in increased systemic, but not ocular, clearance. These properties make RG7716 a potential next-generation therapy for neovascular indications of the eye.

**Keywords** age-related macular degeneration; angiogenesis; angiopoietin-2; Fc receptor; vascular endothelial growth factor
**Subject Categories** Neuroscience; Pharmacology & Drug Discovery; Vascular Biology & Angiogenesis

## Introduction

The retina is supplied with oxygen, nutrients, and waste exchanges by two distinct vascular beds: the retinal and choroidal capillary networks. Abnormal leakage and/or neovascularization from retinal vessels are the hallmarks of diseases such as diabetic macular edema (DME), diabetic retinopathy (DR), and retinal vein occlusion (RVO) (Penn et al, 2008; Wang & Hartnett, 2016). Wet age-related macular degeneration (wAMD) develops when new vessels grow into the subretinal space from the underlying choroidal capillary network underneath the outer retina (Campochiaro, 2013). Micro-angiopathy, neovascularization, and vascular leakage from choroidal and retinal capillaries are the most common causes of moderate and severe vision loss in developed countries (WHO, 2014).

The importance of vascular endothelial growth factor (VEGF)-A in pathological angiogenesis, particularly in oncology and eye diseases, led to the development of several agents for clinical intervention. Anti-VEGF-A-based therapies are now the standard of care for a variety of solid tumor indications and ocular neovascular diseases, such as wAMD, DME, and RVO. One way of enhancing the efficacy of anti-VEGF-A-based therapies is adding the inhibition of another soluble factor to the therapeutic reagent that is also essential for angiogenesis (Jo et al, 2006).

1 Roche Pharma Research and Early Development, Roche Innovation Center München, Penzberg, Germany
2 Department of Ocular Biology and Therapeutics, UCL London, Institute of Ophthalmology, London, UK
3 Roche Pharma Research and Early Development, Roche Innovation Center Basel, F. Hoffmann-La Roche Ltd, Basel, Switzerland
4 Translational Pre-Clinical Model Platform, Singapore Eye Research Institute, The Academia, Singapore, Singapore
5 The Ophthalmology & Visual Sciences Academic Clinical Program, DUKE-NUS Graduate Medical School, Singapore, Singapore
6 Department of Ophthalmology, Goethe University, Frankfurt am Main, Germany
7 Department of Ophthalmology, Ruprecht Karls University, Heidelberg, Germany
8 Roche Pharma Research and Early Development, Roche Innovation Center Zürich, F. Hoffmann-La Roche Ltd, Zürich, Switzerland
*Corresponding author. Tel: +41 616874588; Fax: +41 616880382; E-mail: guido.hartmann@roche.com
†These authors contributed equally to this work

The angiopoietins ANG-1 and ANG-2 are growth factors that play a key role in vessel homeostasis, angiogenesis, and vascular permeability. Both ligands interact with the Tie-2 transmembrane receptor tyrosine kinase. Tie-2 is preferentially expressed on endothelial cells and a subset of myeloid cells. Loss- and gain-of-function experiments have demonstrated the critical contributions of the angiopoietin/Tie-2 system in vascular development and vascular diseases (Puri *et al*, 1995; Asahara *et al*, 1998; Augustin *et al*, 2009). Tie-2 receptor-deficient mice die by embryonic day 10.5 as a result of vessel immaturity and incorrect vessel organization (Dumont *et al*, 1994; Sato *et al*, 1995). ANG-1-deficient mice show a phenotype that is reminiscent of the Tie-2-deficient phenotype (Suri *et al*, 1996), while transgenic overexpression of ANG-2 mimics the phenotype of Tie-2- and ANG-1-deficient mice, suggesting that ANG-1 and ANG-2 have opposing activities. Both Tie-2 ligands bind to the receptor with similar affinity, whereas only ANG-1 induces strong phosphorylation of the kinase domain and subsequent signaling events. ANG-2 has been characterized as a partial Tie-2 agonist that competitively inhibits ANG-1 signaling (Maisonpierre *et al*, 1997; Saharinen *et al*, 2008). Structural studies have identified an agonistic loop (P-domain) in ANG-1 that is capable of converting ANG-2 into a full agonist on chimerization (Yu *et al*, 2013). ANG-2 also amplifies proapoptotic signals in pericytes under stress from elevated glucose conditions or proinflammatory cytokines like TNF-α (Cai *et al*, 2008; Park *et al*, 2014). Overall data suggest that ANG-1 protects from pathological angiogenesis and drives toward a quiescent, mature vessel phenotype (Nambu *et al*, 2004; Hammes *et al*, 2011; Lee *et al*, 2014), whereas ANG-2 promotes vascular leakage and hypotensive, abnormal vessel structure (Ziegler *et al*, 2013).

Interaction of the angiopoietins with the VEGF-A pathway has been studied in ocular setting using double transgenic mice co-expressing ANG-1 and VEGF-A. ANG-1 expression, when initiated simultaneously with VEGF-A, suppressed VEGF-A-induced neovascularization and prevented retinal detachment (Nambu *et al*, 2005). Although ANG-1 effectively blocked the initiation and progression of VEGF-A-induced neovascularization, no impact on previously established lesions was observed (Nambu *et al*, 2005). Conversely, co-expression of ANG-2 with VEGF-A in the developing retina and in ischemic retina models showed accelerated neovascularization

compared to VEGF-A expression alone (Oshima *et al*, 2004, 2005). Further pathophysiological significance arises from reports of elevated levels of VEGF-A and ANG-2 in vitreous samples from diabetic patients undergoing vitrectomy, which correlate with each other and with disease severity (Watanabe *et al*, 2005; Loukovaara *et al*, 2013). Therefore, VEGF-A and ANG-2 appear strongly linked to normal vascular development, but also to pathological neovascularization and vascular permeability, which are the hallmarks of ocular neovascular disease.

Several bispecific antibodies are currently in clinical development, including an anti-VEGF-A/ANG-2 bispecific antibody targeting neovascularization in tumors (Kienast *et al*, 2013). The aim of this study was to explore the potential of dual VEGF-A and ANG-2 inhibition in the management of retinal vascular diseases. We tested monotherapy and dual inhibition of VEGF-A/ANG-2 using a species crossreactive CrossMAb in the spontaneous model of CNV in JR5558 mice. We then developed a human bispecific CrossMAb (RG7716), which binds and neutralizes human VEGF-A and ANG-2 with high potency and tested it in the laser-induced model of CNV in non-human primates. Our data suggest that dual inhibition of these angiogenic signaling circuits improves outcomes in preclinical models of ocular neovascular disease.

# Results

## ANG-2 levels are elevated in human retinal vascular diseases

To investigate whether ANG-2 is a suitable pharmacological target for human retinal vascular diseases, we measured the levels of ANG-1 and ANG-2 in the vitreous of patients newly diagnosed with wet AMD, DR, proliferative DR, or RVO in comparison with macular hole as controls, as described in Koss *et al* (2011). In human vitreous samples, ANG-1 levels were weakly elevated in RVO (71.1 up to 107 pg/ml) and decreased in proliferative DR (down to 36.3 pg/ml) compared to controls (Fig 1A). However, levels of ANG-2 were significantly elevated in all four retinal vascular diseases investigated compared to controls (Fig 1B). From control levels of 68.4 pg/ml, ANG-2 increased to 139 pg/ml in wet AMD, to 302 pg/ml in DR, to 1,140 pg/ml in RVO, and to 1,625 pg/ml in proliferative DR.

**Figure 1.   Vitreous concentrations of angiopoietins in patients newly diagnosed with retinal diseases and cell model of barrier breakdown testing the interaction of VEGF-A and angiopoietins.**

A, B    Box plots of vitreal ANG-1 (A) and ANG-2 (B) levels from newly diagnosed patients with wAMD, DR, proliferative DR and RVO compared to controls (macular hole). The interquartile range of the data is indicated by the box. A nonparametric Kruskal–Wallis analysis followed by Dunn's method for multiple comparisons was used to show significant differences of the groups to control which are indicated by asterisks. ANG-1 levels did not differ significantly, but ANG-2 levels were significantly different: control vs. AMD (*, $P = 0.0451$), vs. DR (****, $P < 0.0001$), pDR (****, $P < 0.0001$), and RVO (****, $P < 0.0001$).

C    Schematic of cellular experiments to measure transendothelial resistance in human endothelial cells using transwell filters and CellZscope technology.

D    24-h concentration of ANG-2 in supernatants of the basal side of the culture stimulated with 5 ng/ml of VEGF-A, 200 ng/ml of ANG-1, or the combination. Error bars show SEM with one-sided ANOVA ($P < 0.0001$) and Tukey's multiple *t*-test for five independent experiments indicating significance for control vs. VEGF-A (**, $P = 0.0028$); VEGF-A vs. ANG-1 (****, $P < 0.0001$); ANG-1 vs. VEGF-A and ANG1 (**, $P = 0.0061$); VEGF-A vs. VEGF-A and ANG-1 (ns, $P = 0.0625$).

E    Human endothelial cells plated on filters were assessed for endothelial barrier function over time after the addition of 5 ng/ml VEGF-A alone or in combination with 200 ng/ml ANG-1 or 10 μg/ml anti-ANG-2 or all three test items. The final time point was used for statistical analysis. Error bars show SEM with one-sided ANOVA ($P < 0.0001$) and Tukey's multiple *t*-test for three independent experiments indicating significance of VEGF-A vs. untreated (****, $P < 0.0001$); vs. VEGF-A and ANG-1 and anti-ANG-2 (***, $P = 0.0004$); vs. VEGF-A and anti-ANG-2 (*, $P = 0.038$); vs. VEGF-A and ANG-1 (*, $P = 0.040$). Furthermore, VEGF-A and anti-ANG-2 and ANG-1 are significantly different vs. VEGF-A and ANG-1 (*, $P = 0.0451$); vs. VEGF-A and anti-ANG-2 (*, $P = 0.0474$); vs. untreated (*, $P = 0.0492$). Finally untreated is significantly different vs. VEGF-A and ANG-1 (***, $P = 0.0009$) and VEGF-A and anti-ANG-2 (***, $P = 0.0008$).

Data information: ANOVA, analysis of variance; SEM, standard error of the mean; ANG-2, angiopoietin-2; DR, diabetic retinopathy; RVO, retinal vein occlusion; wAMD, wet age-related macular degeneration; pDR, proliferative diabetic retinopathy; vs., versus.

### Interplay of VEGF-A and angiopoietins in a model of endothelial barrier breakdown

We then investigated the interplay between VEGF-A and the angiopoietins in a barrier breakdown model in human primary endothelial cells, in order to better understand some of the cellular aspects leading to hyperpermeability in patients (Fig 1C–E). Human endothelial cells in culture secrete large amounts of ANG-2 into the supernatant. VEGF-A increases the amount of ANG-2 even further, whereas ANG-1 reduces the secretion of ANG-2. Combined exposure of endothelial cells to VEGF-A and ANG-1 showed a trend to reduce

ANG-2 production induced by VEGF-A (Fig 1D). Endothelial barrier breakdown is a key event in retinal eye disease, with edema being a major driver of pathology in the retina. VEGF-A is a known inducer of endothelial barrier breakdown and, indeed, we measured progressive loss of endothelial barrier function over time in our model. We then tested whether ANG-2 present in the culture contributes to VEGF-A-induced barrier breakdown. When adding anti-ANG-2 together with VEGF-A, barrier breakdown was reduced. This suggests that VEGF-A, at least partly, signals via ANG-2 to trigger endothelial barrier breakdown. The addition of ANG-1 also reduced VEGF-A-induced barrier breakdown, demonstrating an improved

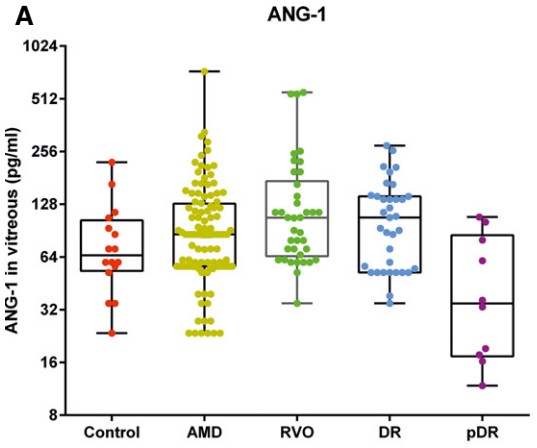

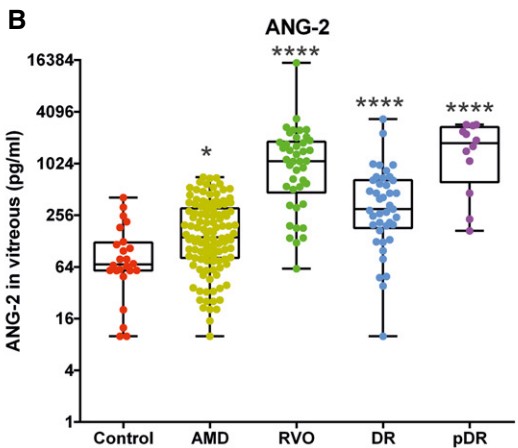

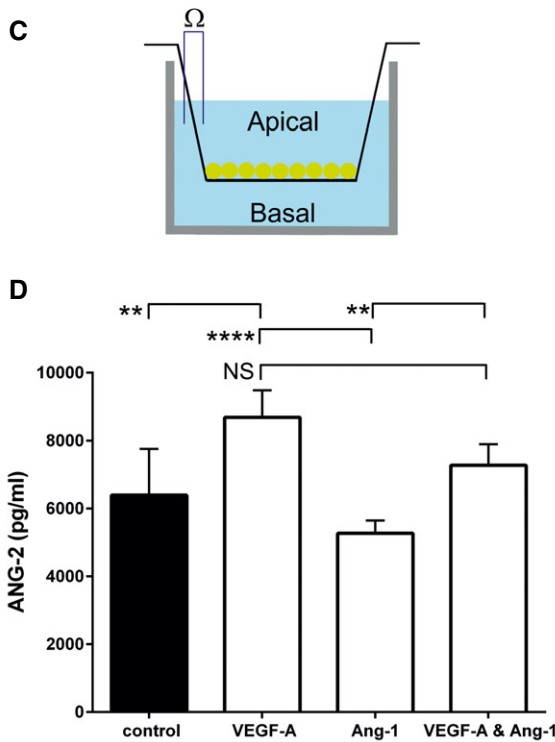

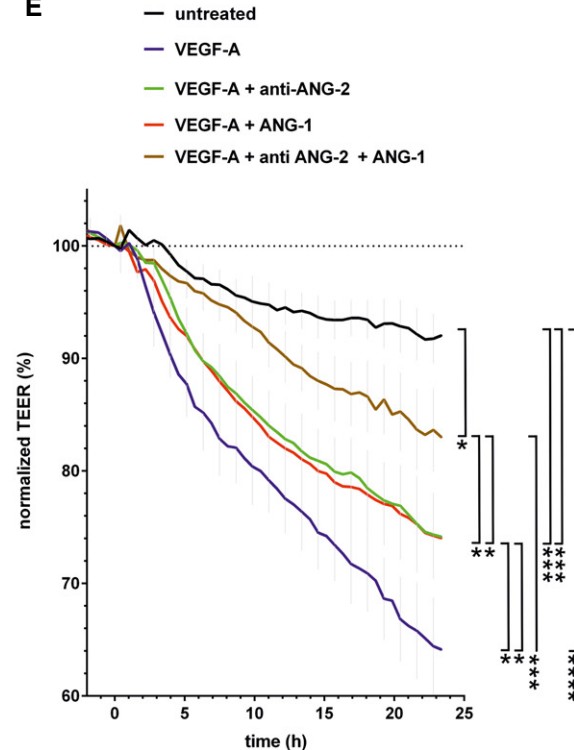

**Figure 1.**

endothelial monolayer integrity function of ANG-1. Combined addition of anti-ANG-2 and ANG-1 reduced the VEGF-A-induced barrier breakdown even further (Fig 1E). The dynamic nature of VEGF-A-induced barrier breakdown was demonstrated when a bispecific anti-VEGF-A/ANG-2 was added 18 h after the addition of VEGF-A. TEER values reverted back to values before the addition of VEGF-A to the culture (Appendix Fig S1). The results confirm the concept of VEGF-A and ANG-2 being drivers of endothelial barrier breakdown, while ANG-1 counteracts these activities and promotes normal vessel integrity.

## Monotherapy of anti-VEGF-A vs. combination therapy of anti-VEGF-A and anti-ANG-2 in rodent models of ocular neoangiogenesis

We then turned to a model of spontaneous CNV (sCNV) in the JR5558 mouse strain to further probe the potential for anti-VEGF-A/ANG-2 combination therapy. The sCNV model develops leaky, neovascular tufts arising from the choriocapillaries, which resemble human choroidal lesions. The pathological consequences of these leaky neovessels distorting the retinal architecture can be seen by increased neuroretinal cell death and loss of retinal functionality observed using electroretinography. The model also shows increased expression of VEGF mRNA in the RPE/choroid and retina when compared to wild-type mice. Inhibition of VEGF receptor-2 using a blocking antibody reduced the severity of neovascularization, making this a suitable model to test for combination therapies that enhance the efficacy of anti-VEGFs (Nagai et al, 2014). We first probed for ANG-2 expression in the model and found that ANG-2 mRNA was elevated in the choroid/RPE, but not retina, of JR5558 mice compared to wild-type mice starting at day 27 and increasing further as measured at days 50 and 62 (Fig 2A and B).

Next, we directly tested the concept of combined inhibition of VEGF-A and ANG-2 being more efficacious than VEGF-A inhibition alone in preventing neovascularization and its associated endothelial dysfunction. In all rodent experiments, we used the rodent crossreactive B20-4.1 Fab arm (Liang et al, 2006) and the anti-ANG2-LC10 antibody generated as CrossMAbs or IgGs, as indicated (Schaefer et al, 2011). Since neovascular tufts arise from the choriocapillaris between postnatal day (D)10 and D15 and lesion numbers peak at approximately D30 in this model (Nagai et al, 2014), administration of antibodies was initiated at D14 and D19 by intraperitoneal (IP) injection (termed early intervention mode; Fig 2C). At D26, fluorescein angiography (FA) was performed, and lesion number and size were evaluated. Anti-VEGF-A and anti-ANG-2 were tested at 5 mg/kg while the anti-VEGF-A/ANG-2 antibody was tested in a dose–response with 10, 5, and 3 mg/kg. A normal IgG$_1$ can bind two molecules of VEGF-A or ANG-2 (one per Fab arm), but an anti-VEGF-A/ANG-2 CrossMAb only one of each ligand. Therefore, the 10 mg/kg high dose of the anti-VEGF-A/ANG-2 CrossMAb has an equal molar concentration of binding sites as 5 mg/kg of standard IgG$_1$ (anti-VEGF-A, anti-ANG-2, or IgG control). Anti-VEGF-A reduced the number and lesion area significantly compared to IgG control, while anti-ANG-2 alone showed a trend toward reduction. The highest doses of anti-VEGF-A/ANG-2 CrossMAb significantly reduced the number and size of lesions compared to IgG control and anti-VEGF-A or anti-ANG-2 monotherapy alone (Fig 2D–K).

Choroidal neovascularization progressively damages the retina, so later time points were investigated in the mouse model to more closely reflect the clinical situation in which a patient presents with established disease. In the late intervention model, antibody is given when a considerable number of lesions with parainflammation and neuronal cell death are already established (Nagai et al, 2014). Antibody was given once a week for 2 weeks from D47 onwards, with examination at D60. In these experiments, we compared both anti-VEGF-A and anti-ANG-2 alone to bispecific VEGF-A/ANG-2 CrossMAb antibody (CrossMAb and IgG control at 3 mg/kg dose,

**Figure 2. Inhibition of vessel leakiness and lesion number by combined inhibition of VEGF-A and ANG-2 in the model of sCNV in JR5558 mice using fluorescence angiography.**

A, B     Real-time qPCR analysis of ANG-2 levels in retina and RPE/choroid complexes of JR5558 and C57BL/6J (C57) mice. (A) Relative expression levels of ANG-2 were significantly increased in the RPE/choroid complexes of JR5558 mice in comparison with C57, at 50 and 62 days old (*, $P = 0.022$ at D50 and *, $P = 0.042$ at D62). (B) By contrast, retina levels of ANG-2 were not significantly different between C57 and JR5558 mice, indicating neovascularization is driven locally by ANG-2. Asterisk (*) denotes statistical significance of JR5558 mice compared to wild-type C57BL/6 using unpaired t-test for each time point analyzed separately.

C     Schematic presentation of experimental design. Mice received IP injections of CrossMAb anti-VEGF-A/ANG-2 (species crossreactive, B20-4.1 and LC10), anti-ANG-2 (LC10), an anti-VEGF-A IgG$_1$ (B20-4.1), or IgG control at postnatal D14 and D19 followed by fluorescence angiography at D26 (early intervention), results demonstrated in (D–K). Alternatively, mice received antibody at postnatal D47 and D55 followed by fluorescence angiography at D60 (late intervention), results demonstrated in (L) and (M).

D–I     Representative examples of fluorescence angiograms of IgG control (D), anti-VEGF-A (E), anti-ANG-2 (F) (all at 5 mg/kg), and three doses of anti-VEGF-A/ANG-2 (G, H, and I) (at 3, 5, and 10 mg/kg).

J, K     Bar graph of numbers of spontaneously occurring lesions (J) and area by fluorescence angiography (K) after two weekly doses of antibody (antibodies at 5 mg/kg IP and anti-VEGF-A/ANG-2 at 10 [high], 5 [mid], and 3 [low] mg/kg IP) followed by analysis a week later in the early intervention model.

L, M     Bar graph of numbers of spontaneously occurring lesions (L) and area by fluorescence angiography (M) after two weekly doses of antibody (3 mg/kg IP) followed by analysis a week later in the late intervention model.

Data information: SEM is shown as error bars with $n = 4$ (A, B) or $n = 8$ (J–M) animals per group and significance indicated by * using ANOVA (J, K: $P < 0.0001$; L: $P < 0.0018$; M: $P < 0.031$ followed by Tukey's multiple t-test in J–M). In (J), IgG control is significant against anti-VEGF-A (****, $P < 0.0001$), anti-ANG-2 (**, $P = 0.0069$), anti-VEGF-A/ANG-2 low (*, $P = 0.0194$, mid and high (****, $P < 0.0001$). Furthermore, anti-VEGF-A/ANG-2 high is significant against anti-VEGF-A (*, $P = 0.0428$), anti-ANG-2 (****, $P < 0.0001$), and anti-VEGF-A/ANG-2 low (****, $P < 0.0001$). Finally, anti-ANG-2 is significantly different from anti-VEGF-A/ANG-2 mid (**, $P < 0.0041$). In (K), IgG control is significant against anti-VEGF-A (***, $P = 0.0003$) and anti-VEGF-A/ANG-2 mid and high (both ****, $P < 0.0001$). Furthermore, anti-VEGF-A/ANG-2 high is significantly different from anti-VEGF-A (**, $P = 0.0037$), anti-ANG-2 (****, $P < 0.0001$), and anti-VEGF-A/ANG-2 low (***, $P = 0.0001$). Finally, anti-ANG-2 is significantly different from anti-VEGF-A/ANG-2 mid (**, $P < 0.0022$) and anti-VEGF-A/ANG-2 low against mid (*, $P = 0.022$). In (L), IgG control is significantly different from anti-ANG-2 (*, $P = 0.023$) and anti-VEGF-A/ANG-2 (*, $P = 0.014$). Vehicle is different from anti-ANG-2 (*, $P = 0.024$) and anti-VEGF-A/ANG-2 (*, $P = 0.016$). In (M), IgG control is significantly different from anti-ANG-2 (*, $P = 0.044$) and anti-VEGF-A/ANG-2 (*, $P = 0.049$). D, day; IP, intraperitoneal; sCNV, spontaneous choroidal neovascularization.

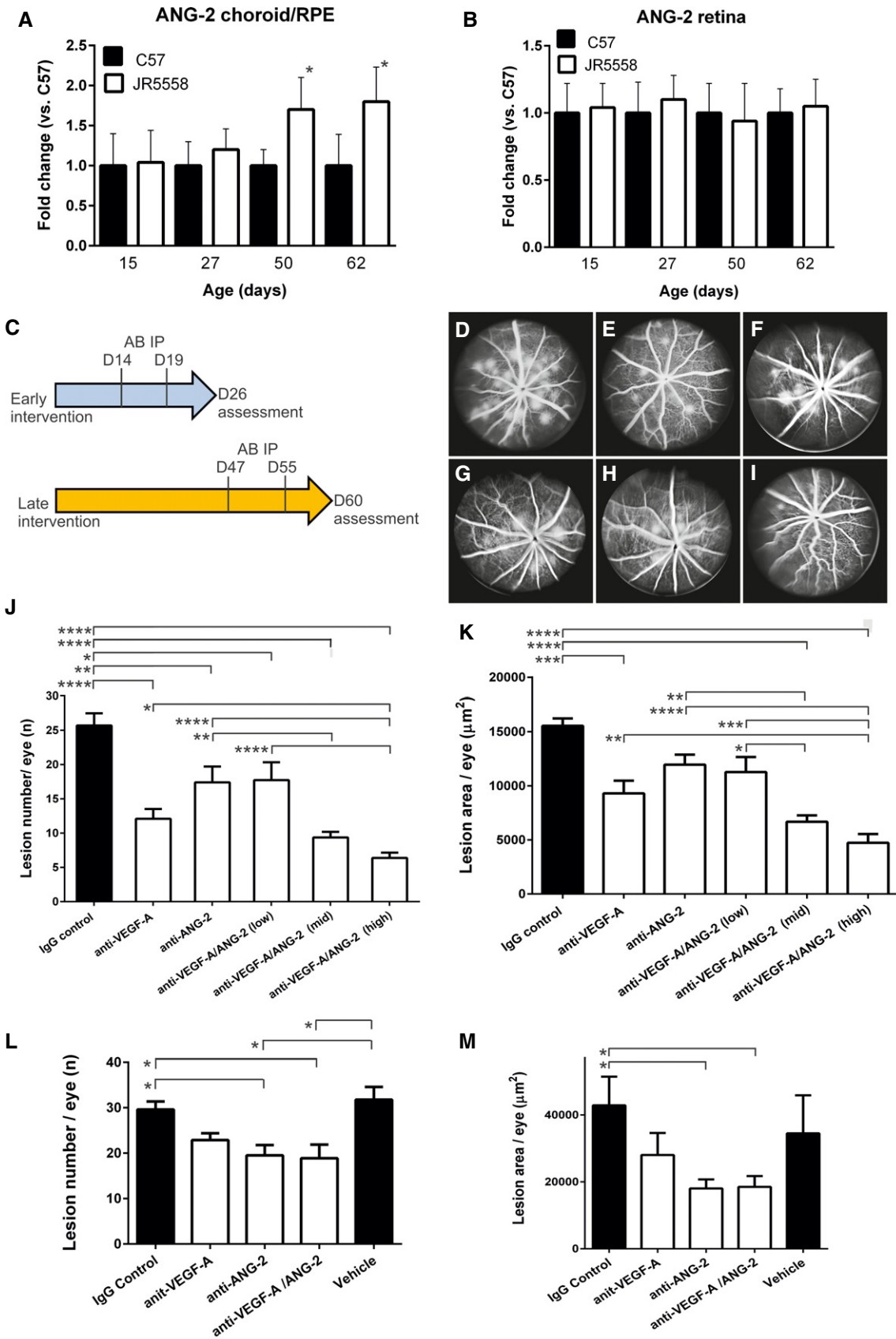

Figure 2.

VEGF-A and ANG-2 antibodies at 1.5 mg/kg). The reduction in the number of lesions with this late intervention regimen was less compared to early intervention when antibodies were administered while lesions were developing (Fig 2L). However, despite a small reduction in lesion number by anti-ANG-2 and anti-VEGF-A/ANG-2, the reduction did reach statistical significance compared to IgG control and vehicle (Fig 2L). In contrast to lesion number, pronounced reductions in the area of lesions were apparent for anti-VEGF-A/ANG-2 CrossMAb compared to IgG control and a trend compared to vehicle under these experimental conditions, as for anti-ANG-2, whereas anti-VEGF-A did not have a statistically significant impact (Fig 2M).

Leaky neovessels in the JR5558 model result in the accumulation of fluid in the retina, which can be monitored noninvasively using optical coherence tomography (OCT). Topographical maps of OCT volumes demonstrated areas of subretinal elevation delineating fluid-filled regions. These areas correlated with regions of leakiness demonstrated by FA (Fig 3B). Image analysis of these areas revealed a reduction in volume of subretinal edema in mice treated with 1, 5, and 10 mg/kg bispecific anti-VEGF-A/ANG-2. No significant reduction was seen with anti-ANG-2 or VEGF-A alone at 5 mg/kg. (Fig 3A and B).

We then tested the concept of combined inhibition of VEGF-A and ANG-2 in pathological neovascularization in a second mouse model, namely oxygen-induced retinopathy (Appendix Fig S2A). It is used as a model mimicking retinal neovascularization in retinopathy of prematurity (ROP) and other vasculopathologies. The VEGF-A dependence of this model has been demonstrated (Aguilar *et al*, 2008; Hartnett *et al*, 2008; Barnett *et al*, 2010). Indeed, anti-VEGF-A treatment at 10 mg/kg IP at D14 increased the avascular area in the center of the retina as anticipated. Dual inhibition with bispecific anti-VEGF-A/ANG-2 increased the avascular area significantly further, beyond IgG control and anti-VEGF-A (Appendix Fig S2B). Significant reduction in neovascular tufts at the periphery was only achieved by the bispecific anti-VEGF-A/ANG-2 treatment under these conditions (Appendix Fig S2C).

### Combined inhibition of VEGF-A and ANG-2 reduces retinal inflammation

Inflammation around the CNV lesion is an important contributory factor in the spontaneous CNV model, and ANG-2 has been described as an enhancer of inflammatory signals in the context of endothelial dysfunction (Fiedler *et al*, 2006; Benest *et al*, 2013). Confocal microscopy was used to assess the number of IBA-1-positive macrophages within and around the CNV lesion that reside proximal to the isolectin-B4-positive endothelial cells around the CNV

lesions (Fig 4A–G). Administration of bispecific anti-VEGF-A/ANG-2 at 10, 5, and 3 mg/kg strongly reduced the amount of IBA-1-positive macrophages in the retina and around the lesions compared to anti-VEGF-A or anti-ANG-2 treatment alone (Fig 4A).

To differentiate whether the reduction in lesion-associated macrophages is a direct anti-inflammatory effect or a consequence of reduced endothelial dysfunction and neovascularization, we tested the antibodies in a proinflammatory mediator challenge model for retinal leukocyte infiltration. Mice were challenged systemically with lipopolysaccharide (LPS). Retinal cross sections and aqueous fluids were then analyzed for leukocyte infiltration. Antibody was given IP at 10 mg/kg 24 h before the LPS challenge. Dual inhibition of VEGF-A and ANG-2 using a CrossMAb significantly reduced the number of leukocytes in the retina and the aqueous compartment (Appendix Fig S3A–C). Administration of anti-VEGF-A and anti-ANG-2 as mixed IgG combinations, rather than bispecific CrossMAb, gave similar results, and demonstrates that the CrossMAb format does not affect the neutralization function of the ligand-binding arm. No single treatment (anti-VEGF-A or anti-ANG2) was as effective as the combination treatments (Appendix Fig S3A–C).

### Combined inhibition of VEGF-A and ANG-2 decreases retinal neuroglial apoptosis and increases visual function in spontaneous CNV model

Invasion of choroid capillaries into the RPE and subretinal spaces, and the associated inflammatory cells, eventually causes disruption of retinal architecture, leading to photoreceptor cell death and dysfunction. We investigated the number of TUNEL-positive cells in the outer retina (photoreceptor nuclei) in the late intervention regimen (Fig 5A–G). Monotreatments (anti-VEGF-A, anti-ANG-2 [both at 5 mg/kg]) and the lowest dose (1 mg/kg) of bispecific VEGF-A/ANG-2 antibody led to a mild reduction in TUNEL-positive cells in the outer retina. In contrast, 5 and 10 mg/kg of the bispecific anti-VEGF-A/ANG-2 significantly reduced numbers by up to 66% compared to controls.

Using electroretinography (ERG), we also assessed neuroretinal function in response to large-field flash stimuli under both scotopic and photopic conditions. A trend toward an increase in photopic B-wave amplitude and reduced latency was observed in animals administered anti-VEGF-A or anti-ANG-2; however, only anti-VEGF-A/ANG-2 had a significant effect compared to the IgG control, under both scotopic and photopic conditions (Fig 5H and I).

Overall, the data indicated that simultaneous inhibition of VEGF-A and ANG-2 efficiently inhibited CNV lesion number and

**Figure 3. Reduction in edema by dual VEGF-A and ANG-2 inhibition in late interference model in the retina of JR5558 mice at D60 as revealed by OCT.**

A   Bar graph showing the relative change in OCT-measured volume between the outer nuclear layer and the RPE layer following intervention with test item. Anti-VEGF-A and anti-ANG-2 were given at 5 mg/kg and anti-VEGF-A/ANG-2 at 1, 5, and 10 mg/kg (low, mid, and high dose).

B   Representative OCT images converted to heat maps with the scaling grade given on the right of the figure delineating fluid-filled areas of the retina. Baseline (D46) and 14-day treatments of IgG control, anti-VEGF-A, anti-ANG-2, and bispecific anti-VEGF-A/ANG-2 at the end of the experiment (D60) are shown. For the region of interest, the corresponding FA image at D60 is also shown as comparator.

Data information: SEM is shown as error bars with *n* = 6 animals per group; significance is shown by * using one-sided ANOVA (*P* = 0.0048) followed by Tukey's multiple *t*-test. Significant changes are shown with anti-VEGF-A/ANG-2 high vs. anti-ANG-2 (**, *P* = 0.0069) and IgG control (*, *P* = 0.018). In addition, the comparison of anti-VEGF-A vs. anti-VEGF-A/ANG-2 (high) is approaching significance (ns, *P* = 0.072). FA, fluorescein angiography; OCT, optical coherence tomography; RPE, retinal pigment epithelium; vs., versus.

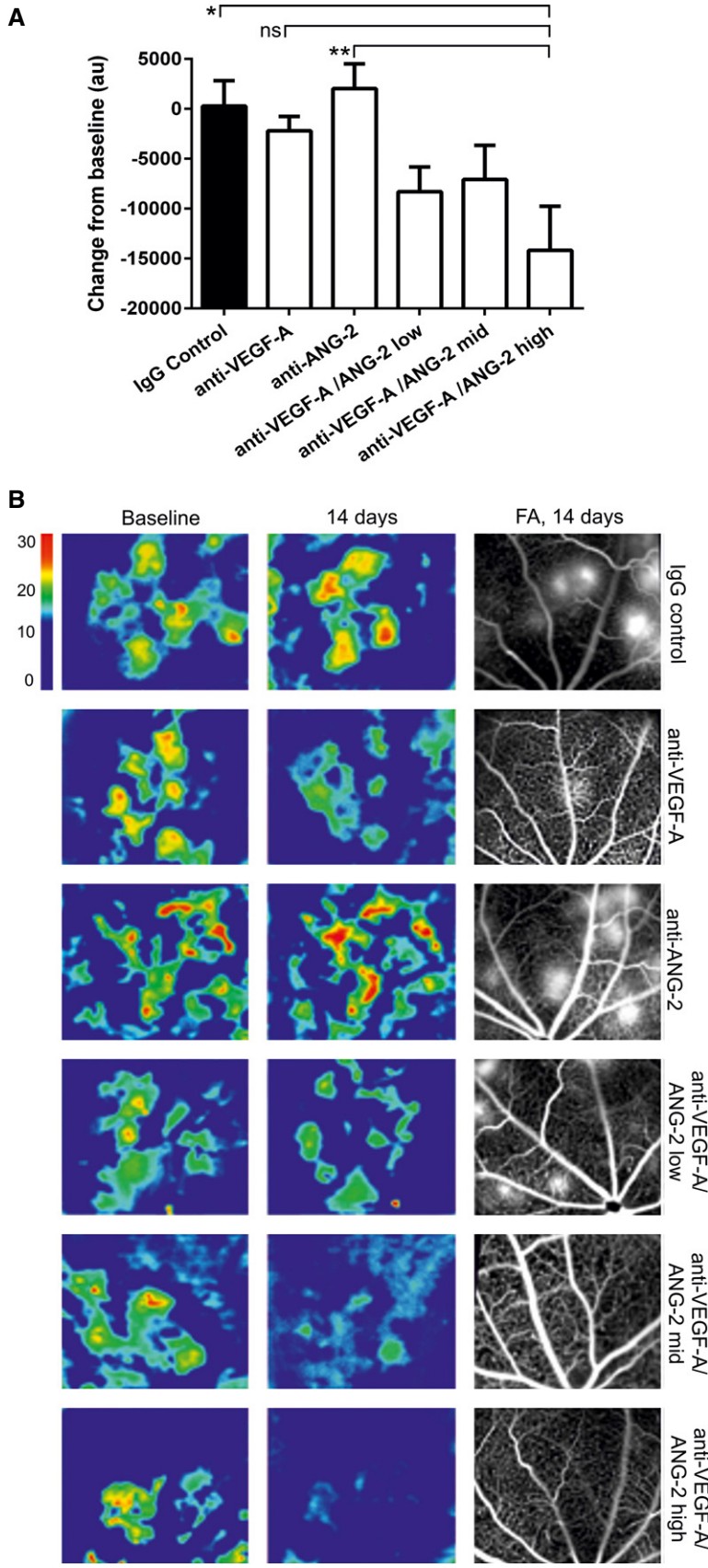

**Figure 3.**

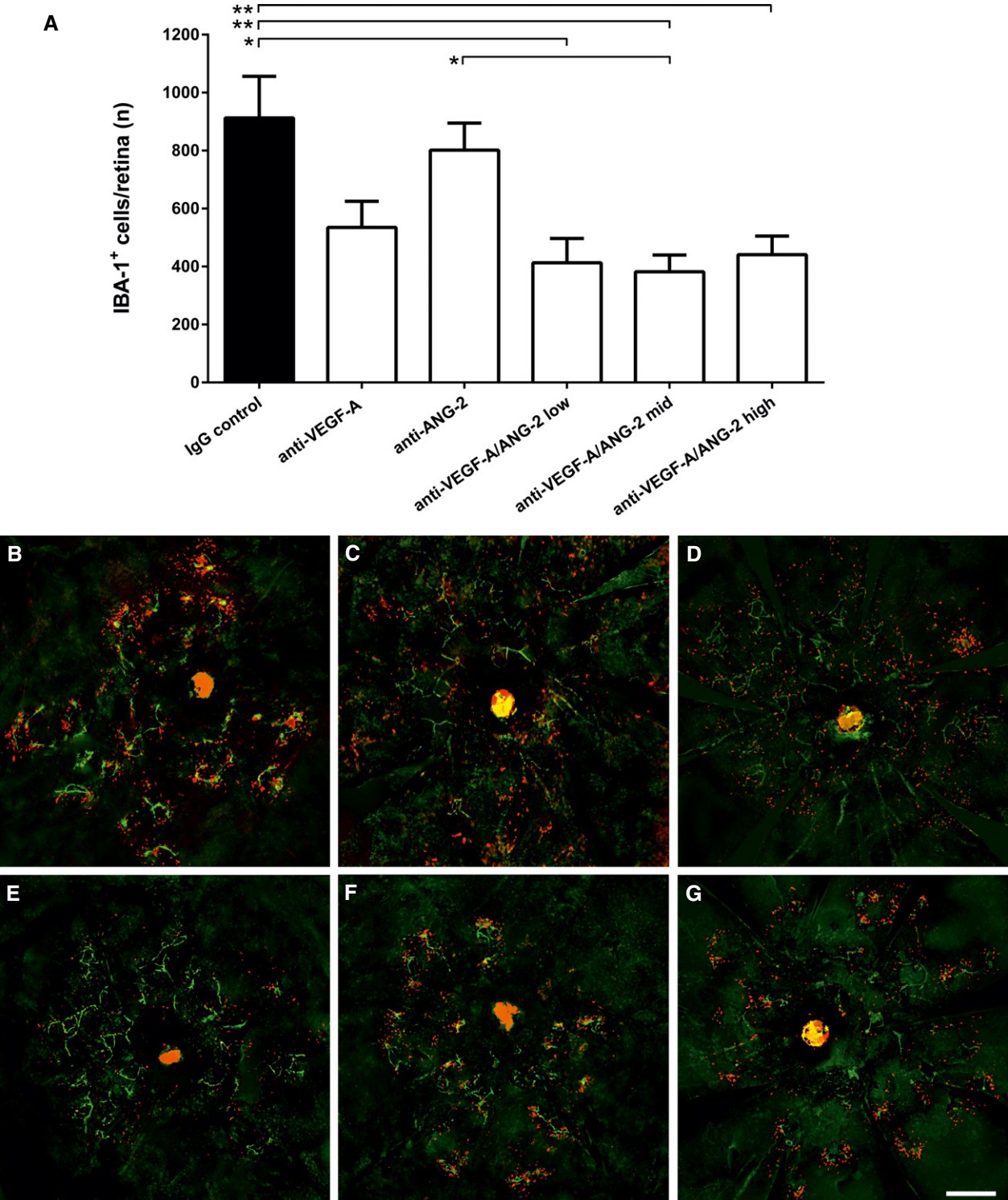

**Figure 4. Dual inhibition of VEGF-A and ANG-2 reduced the number of IBA-1+ macrophages in the model of sCNV in JR5558 mice.**

A    Bar graph of number of IBA-1+ cells in the retina of mice treated with IgG control, anti-VEGF-A, anti-ANG-2 (all 5 mg/kg IP), or bispecific anti-VEGF-A/ANG-2 (at 10, 5, and 3 mg/kg) in the late interference model at D60. Error bars show SEM with *n* = 6 animals per group. All significant differences are shown by * after one-sided ANOVA (*P* < 0.0008) and Tukey's multiple *t*-test. IgG control is significantly different vs. anti-VEGF-A/ANG-2 low (*, *P* < 0.0134), mid (**, *P* < 0.0078), and high (**, *P* < 0.0042); furthermore, anti-ANG-2 is significantly different vs. anti-VEGF-A/ANG-2 mid (*, *P* < 0.0353).

B–G   (B) IgG control, (C) anti-VEGF-A, (D) anti-ANG-2, (E–G) anti-VEGF-A/ANG-2 low, mid, and high: representative micrographs of whole-mount eyecup preparations stained with isolectin-B4 positive (green) to stain vessels and anti-IBA1 (red) to stain for macrophages show a decrease in both absolute numbers and clustering of subretinal IBA-1-expressing cells associated with neovascular complexes in anti-VEGF-A/ANG-2 treatment groups. Scale bar, 200 μm.

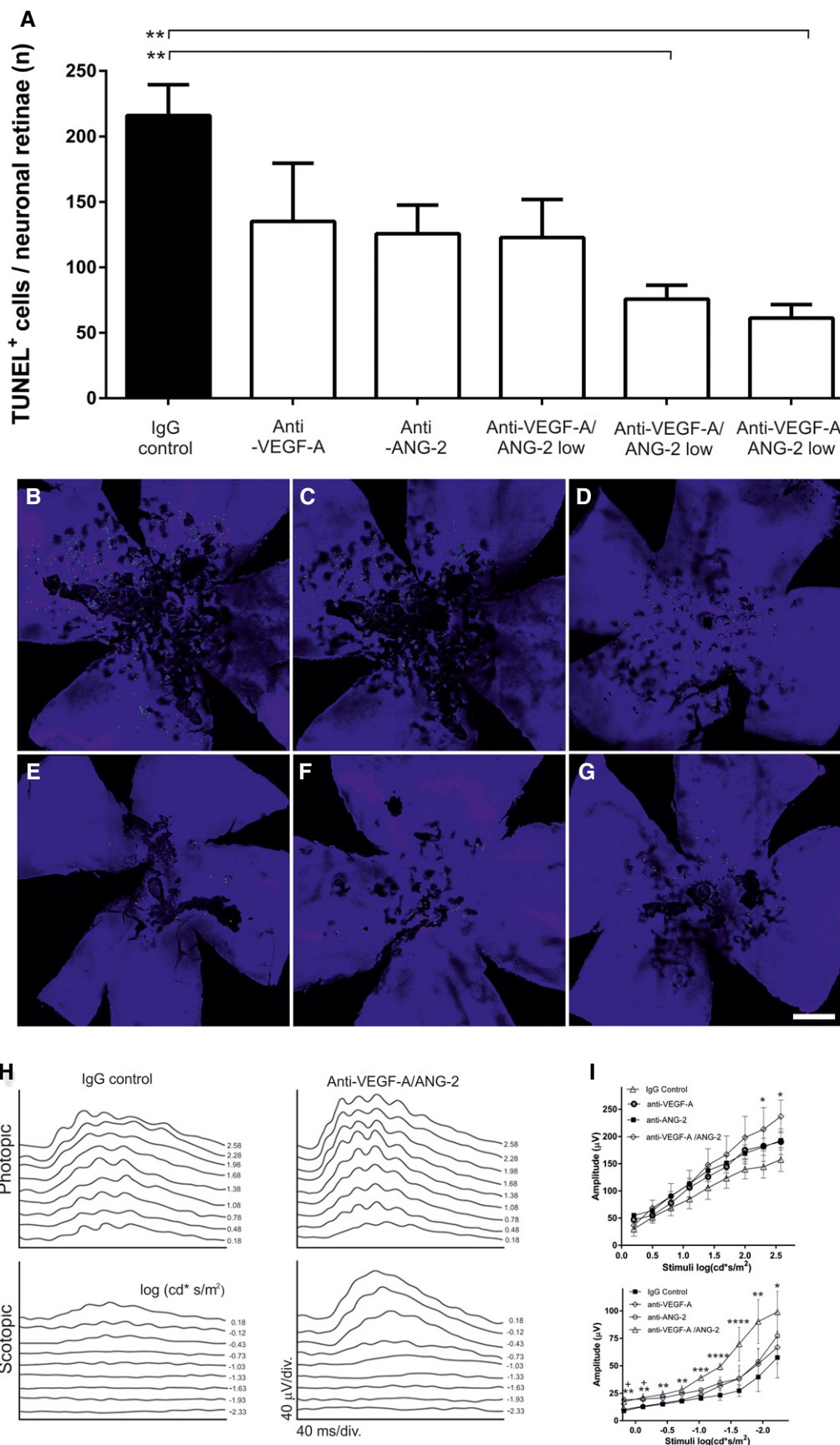

**Figure 5.**

◀

**Figure 5.　Dual inhibition of VEGF-A and ANG-2 reduced the number of TUNEL-positive cells and improved visual functionality, as revealed by ERG, in a model of sCNV in JR5558 mice.**

A　Bar graph of TUNEL-positive cells in the retina of mice treated with 5 mg/kg anti-VEGF-A, anti-ANG-2, and 3, 5, and 10 mg/kg anti-VEGF-A/ANG-2. Error bars show SEM with $n = 6$ animals per group and * denotes all significant changes after one-sided ANOVA ($P = 0.0041$) and Tukey's multiple $t$-test. IgG control is significantly different from anti-VEGF-A/ANG-2 mid (**, $P = 0.0077$) and high (**, $P = 0.0028$).

B–G　(B) IgG control, (C) anti-VEGF-A, (D) anti-ANG-2, (E–G) anti-VEGF-A/ANG-2 low, mid, and high: representative micrographs of whole-mount retinae preparations display the reduction in TUNEL-positive photoreceptor cells (green) clustered around focal neovascular lesions in anti-VEGF-A/ANG-2 (E–G) and, to a lesser extent, anti-VEGF-A-treated (C) and anti-ANG-2-treated (D) groups compared with IgG-treated controls (B) (scale bar, 200 μm).

H　ERG was used to assess retinal function in response to large-field flash stimuli under both scotopic and photopic conditions. Representative photopic (light-adapted) and scotopic (dark-adapted) flash response series show increased amplitude and reduced latency after treatment with bispecific anti-VEGF-A/ANG-2 compared with IgG control (both 3 mg/kg). ERG shows reduced depression of B-wave responses in JR5558 mice treated with anti-VEGF-A/ANG-2 compared to IgG control.

I　Maximum photopic ERG amplitude was increased using anti-VEGF-A/ANG-2 compared to single anti-VEGF-A or anti-ANG-2 treatments (all at 3 mg/kg). There was a significant increase in amplitude at stimuli > 2 log (cd*s/m$^2$) between the bispecific anti-VEGF-A/ANG-2 and IgG control group under scoptopic and photopic conditions. Asterisk (*) denotes significance after ANOVA and Dunnett's multiple $t$-test compared to IgG control compared for each stimuli separately; error bars show SEM with $n = 8$ animals per group. Under photopic conditions, anti-VEGF-A/ANG-2 reached significance at 2.58 and 2.30 cd*s/m$^2$ with *, $P = 0.0498$ and *, $P = 0.0479$, respectively. Under scotopic conditions, all stimuli reached significance for anti-VEGF-A/ANG-2 at −2.3 cd*s/m$^2$ (*, $P = 0.0498$); −1.93 cd*s/m$^2$ (**, $P = 0.0035$); −1.63 and −1.33 cd*s/m$^2$ (****, $P < 0.0001$); −1.03 cd*s/m$^2$ (***, $P = 0.0002$); −0.73 cd*s/m$^2$, (**, $P = 0.004$); −0.43 cd*s/m$^2$, (**, $P = 0.0024$); −0.13 cd*s/m$^2$ (**, $P = 0.0024$) and 0.18 cd*s/m$^2$ (**, $P = 0.0031$). In addition, anti-ANG-2 treatment reached significance at −0.13 cd*s/m$^2$ (+$P = 0.011$) and 0.18 cd*s/m$^2$ (+$P = 0.005$), respectively.

Data information: ERG, electroretinogram; sCNV spontaneous choroidal neovascularization.

---

area, retinal edema, and inflammation, plus promoted the survival and functionality of the neuroretina, and thus could be a promising strategy for the treatment of patients with ocular neovascular disease.

**Generation and characterization of a human CrossMAb antibody targeting VEGF-A and ANG-2 with a modified Fc region optimized for use in ophthalmology**

*High-affinity binding to VEGF-A and ANG-2 but no binding to ANG-1*

To produce a molecule suitable for clinical development, we generated the bispecific CrossMAb RG7716 based on the framework of a 150-kDa human IgG$_1$ (Schaefer *et al*, 2011). RG7716 comprises two different heavy chains and two different light chains. One ligand-binding arm binds VEGF-A and the other binds ANG-2 (Fig 6A). The affinity of the ligand binding of RG7716 was determined using isothermal titration calorimetry (ITC), which offers the advantage of affinity measurements in solution without the strong influence of avidity. RG7716 bound human VEGF-A$_{165}$ and VEGF-A$_{121}$ with K$_D$ values of 3 nM, which was comparable to the K$_D$ of ranibizumab (Fig 6B). Angiopoietins are oligomeric proteins, which are difficult to use in kinetic studies. We therefore used the angiopoietin receptor-binding domains (RBDs) as a Fc fusion protein (Barton *et al*, 2006). RG7716 bound human ANG-2 RBD-Fc fusion protein with a K$_D$ value of 22 nM, whereas no binding of human ANG-1 RBD-Fc fusion protein was detected (Fig 6B). To confirm that RG7716 has no impact on ANG-1 signaling, we compared LC10 (ANG-2 binding part in RG7716) to LC08, a dual ANG-1/ANG-2 binding antibody, in ANG-1-mediated endothelial survival assays using 120 ng/ml ANG-1 as a stimulus. We tested concentrations of the antibodies up to 100 μg/ml and observed a clear dose-dependent inhibition using the dual binding antibody LC08; however, LC10 did not inhibit ANG-1-mediated endothelial survival at any doses tested (Fig 6C). Independent and simultaneous binding of both ANG-2 and VEGF-A by RG7716 was demonstrated in surface plasmon resonance (SPR) experiments (Fig 6D).

*Optimization of RG7716 for retinal indications by engineering of the Fc region*

To optimize RG7716 for ophthalmological use, the Fc region was engineered to abolish binding interactions with all FcγR and FcRn (Fig 7A–E, Table 1). Exchange of amino acids required for the FcγR interactions should eradicate effector functions including antibody-dependent cytotoxicity (ADCC), antibody-dependent cell phagocytosis (ADCP), and complement-dependent cytotoxicity (CDC). Eliminating the FcRn binding site should reduce the systemic half-life compared to wild-type IgG (due to the lack of IgG recycling). The individual point mutations are listed in Table 2. To demonstrate loss of binding of RG7716 to the appropriate human Fc receptors, FcγRIa, FcγRIIa, FcγRIIIa, and FcRn proteins were immobilized on a SPR affinity chip. No binding of RG7716 to Fc receptors was detected, in contrast to the appropriate wild-type and control anti-VEGF-A/ANG-2 antibodies (Fig 7A–E). The mode of action of RG7716 is based on ligand binding and neutralization; hence, the effector function of RG7716 is not needed. There are reports that antibodies with two VEGF-A-binding domains are capable of activating platelets in the presence of proteoglycans, resulting in degranulation and thrombosis *in vivo* (Meyer *et al*, 2009; Julien *et al*, 2014). We therefore tested whether RG7716 has a similar potential to activate platelets. Freshly isolated human platelets were activated in the presence of complexes of heparin, VEGF-A, and RG7716 or anti-VEGF-A. Although an anti-VEGF-A antibody with two antigen-binding sites was capable of activating platelets, no effect was seen with RG7716 (Fig 8A). The effect of eliminating the FcRn binding site in RG7716 was tested in a comparative pharmacokinetic study in non-human primates. We observed faster systemic clearance and lower systemic exposure of RG7716 compared to anti-VEGF/ANG-2-FcγR− (CrossMAb with intact FcRn binding) after single intravitreal administration in cynomolgus monkeys (*Macaca fascicularis*); while systemic concentrations were reduced, the concentration of both drugs in aqueous humor was similar (Fig 8B and C). In vitreous humor, concentrations were measured only for RG7716. The concentrations in the vitreous

**A**

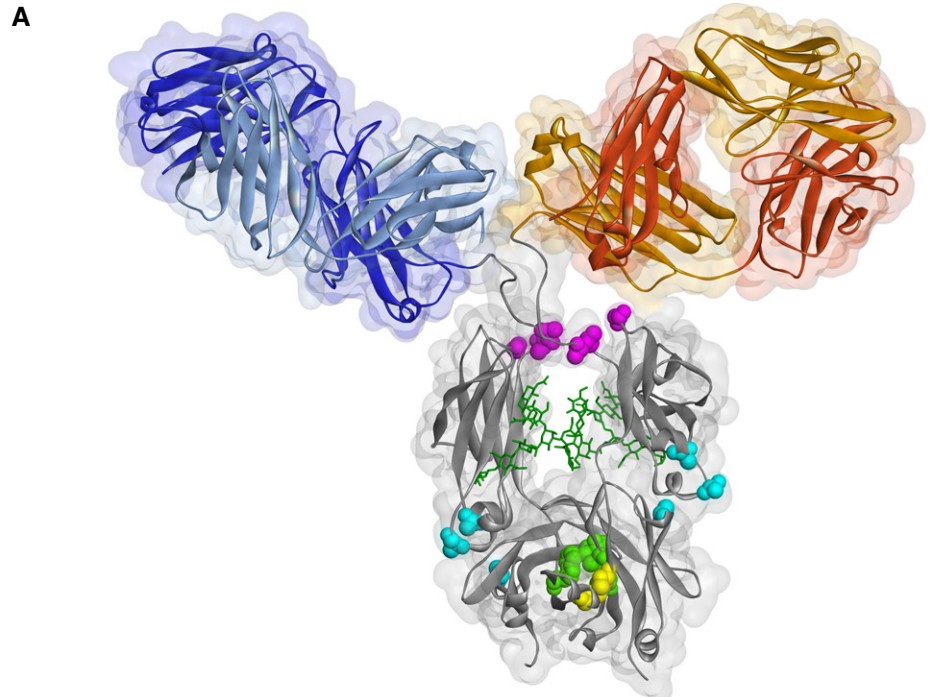

**B**

| Affinity Binding $K_d$ (nM) | Anti-VEGF-A (ranibizumab) | Anti-ANG-2 (LC10) | RG7716 |
|---|---|---|---|
| VEGF-A$_{121}$ | 3.1 | No binding | 3.3 |
| VEGF-A$_{165}$ | 2.2 | No binding | 3.5 |
| ANG-2-RBD-Fc Dimer | No binding | 4.1 | 22 |
| ANG-1-RBD-Fc Dimer | No binding | No binding | No binding |

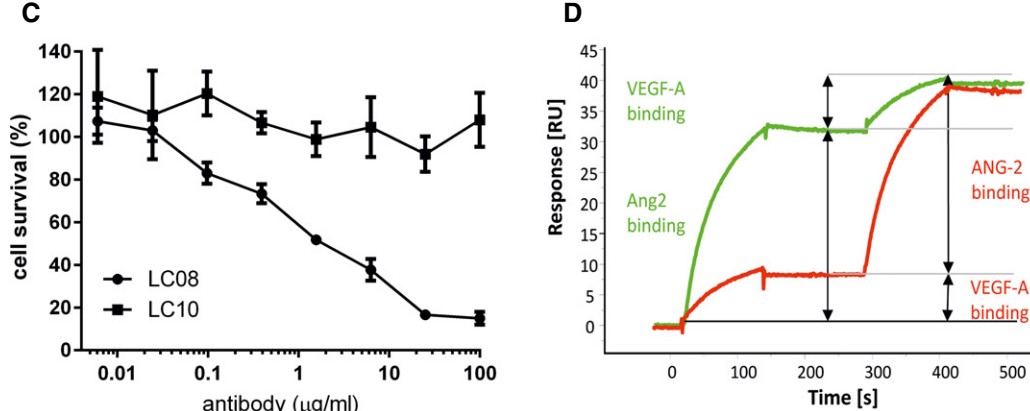

**Figure 6.  Structural characteristics and functional binding properties of CrossMAb RG7716.**

A   Structural presentation of CrossMAb RG7716 with substituted amino acids highlighted and colored. Amino acids corresponding to individual point mutations ensuring correct and efficient heavy chain heterodimerization ("knobs-into-holes" [green] and additional disulfide bridge [yellow]), abolishing Fc receptor functionality (Fcγ receptors I, II, and III, [pink] and FcRn [blue]) are highlighted.

B   Summary table of experimental binding affinities of the parental components targeting VEGF-A (ranibizumab) and ANG-2 (LC10) as well as of RG7716 measured by isothermal titration calorimetry. RG7716 binds VEGF-A$_{121}$ and VEGF-A$_{165}$ with an affinity comparable to ranibizumab (3.3 vs. 3.1 nM). ANG-2 binding of RG7716 is slightly decreased to 22 nM compared to α-ANG-2 IgG$_1$ (LC10) with 4.1 nM. No binding to ANG-1 was detectable for RG7716 or α-ANG-2 IgG$_1$ (LC10).

C   Inhibition of ANG-1-induced endothelial cell survival was tested in the presence of increasing concentrations of LC10 (anti-ANG-2 IgG, ANG-2 binding part in RG7716) and LC08 (ANG-1 and ANG-2 binding antibody). An average of three independent experiments is shown with SEM.

D   The ability of RG7716 to bind to ANG-2 and VEGF-A simultaneously and independent is demonstrated by a surface plasmon resonance experiment. RG7716 is captured via an anti-Fc antibody on the chip surface. The green curve depicts the experiment with ANG-2 in the first and VEGF-A in the second binding event. The red curve shows the binding events in the other sequence.

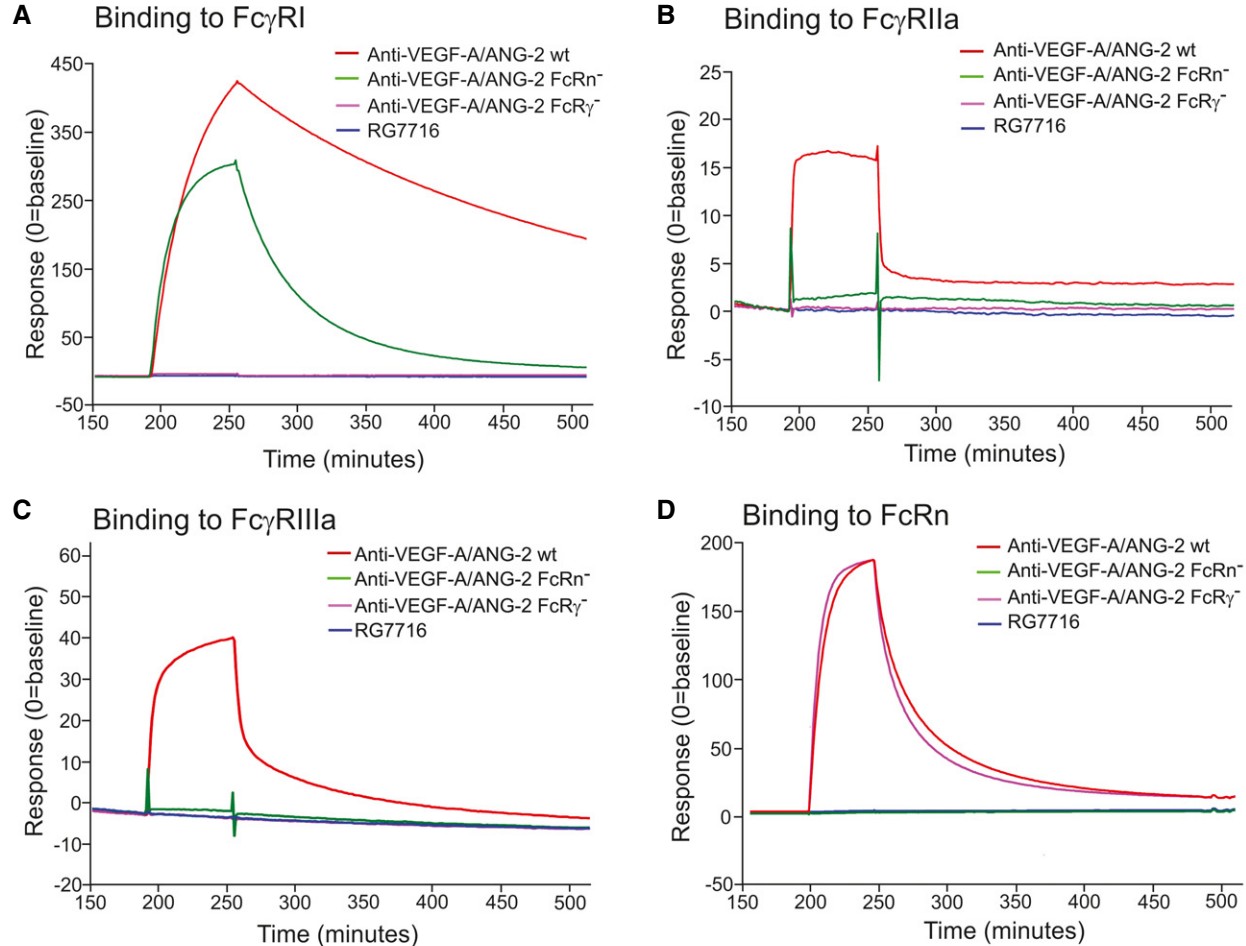

**Figure 7. Binding of CrossMAbs with modification of the Fc part to FcγRI, II, and III, as well as to FcRn in SPR interaction assays.**

A–D   Time-dependent binding of RG7716 is shown in blue, anti-VEGF-A/ANG-2 without modification of the Fc part in red (wild-type), anti-VEGF-A/ANG-2 with modification of the FcRn binding site only in green (FcRn⁻), and anti-VEGF-A/ANG-2 with modification of the FcγR binding in pink (FcγR⁻). Panel (A) shows SPR binding profile over time to immobilized FcγRI, (B) to immobilized FcγRII, (C) to immobilized FcγRIII, and (D) to immobilized FcRn. SPR, surface plasmon resonance.

E     Summary data table of binding data for RG7716 in comparison with anti-VEGF-A/ANG-2 CrossMAbs without modifications in the Fc part.

paralleled those in the aqueous over time, with the area under the curve being approximately four times higher for the vitreous. Flip-flop pharmacokinetics was observed for RG7716, with a strong correlation between exposure in vitreous humor, aqueous humor, and systemic circulation. These properties of RG7716 with good ocular, but low systemic, exposure were also confirmed in the following laser CNV study in non-human primates (Appendix Fig S4A and B).

**Table 1. Schematic presentation of amino acid changes of the Fc part introduced into the human IgG$_1$ framework of a CrossMAb.**

| Antibody and controls/ Fc mutations | wt | PG LALA | AAA (IHH) |
|---|---|---|---|
| Wild-type anti-VEGF-A/ANG-2 | y | – | – |
| Anti-VEGF-A/ANG-2-FcRγ⁻ | – | y | – |
| Anti-VEGF-A/ANG-2-FcRn⁻ | – | – | y |
| RG7716 | – | y | y |

Wild-type anti-VEGF-A/ANG-2 shows naturally occurring configuration while anti-VEGF-A/ANG-2 FcRγ⁻ and anti-VEGF-A/ANG-2 FcRn⁻ have three amino acid substitutions that disable FcγR or FcRn binding, respectively. RG7716 is carrying all amino acid substitutions disabling binding to both FcR classes.

## Direct comparison of RG7716 to ranibizumab in the laser-induced CNV model in non-human primates

*Increased efficacy of RG7716 on reducing choroidal neovascularization*
The laser-induced CNV model in non-human primates was chosen to compare the efficacy of RG7716 to the anti-VEGF-A Fab fragment ranibizumab, a US Food and Drug Administration (FDA)-approved therapy for wAMD since 2006. In the experimental setup, the antibodies were delivered by intravitreal injection, as done in the clinical setting. The model offers the advantage of the similarities in the cynomolgus monkey and human eyes, and circumvents the weak inhibitory activity of ranibizumab against rodent VEGF-A. Lesions were allowed to develop for 14 days following the initial laser injury. A baseline fluorescein angiogram was taken from each eye at D14 to determine the degree of vessel leakiness prior to antibody administration. Lesions were blind-scored on a scale of 1–4, with 4 representing bright hyperfluorescence with late leakage beyond the laser spot, 3 representing hyperfluorescence with late leak only, 2 representing hyperfluorescence only with no leakage, and 1 representing no hyperfluorescence or leakiness. The average score per eye over all groups was 3.45 at D14. Antibodies were delivered by intravitreal injection on D15, 1 day after the baseline fluorescein angiogram (Fig 9A). Antibody was injected at equal molar concentration of anti-VEGF-A binding sites with 90 μg of 150-kDa RG7716 being equal to 30 μg of 50-kDa anti-VEGF-A Fab fragment (ranibizumab). A lower dose of RG7716 (30 μg), anti-ANG-2 (90 μg), and an isotype control (90 μg) was also tested. At D28, a second fluorescein angiogram determined the change in vessel leakiness from baseline (at D14). Lesions from control IgG antibody-treated eyes remained leaky and showed no change in the degree of vessel leakiness from baseline. All other antibodies reduced the degree of severity compared to controls. Ranibizumab reduced the severity score by 0.85 and anti-ANG-2 by 0.57. The 30 μg dose of RG7716 reduced the score by 0.67 and the 90 μg dose by 1.35. The equimolar dose of VEGF-A binding sites for RG7716 (90 μg) was significantly better than anti-VEGF-A (ranibizumab) in reducing vessel leakiness (Fig 9B–G). The degree of vessel leakiness was associated with morphological changes in the retina at the site of laser injury. A distorted retinal architecture was apparent if a laser lesion was control-treated at the D28 time point, as revealed by thickening of the retina, fibrosis and edematous vacuoles; this was reduced upon treatment with RG7716 (Fig 9H–L).

*Reduction in VEGF-A, ANG-2, and proinflammatory cytokines in aqueous humor of antibody-treated animals*
Levels of proangiogenic factors (VEGF-A, ANG-2) and proinflammatory cytokines interleukin-6 and interleukin-8 (IL-6 and IL-8) in aqueous were compared in the treatment groups to the isotype control. A first sample was taken 1 day after intravitreal delivery of the antibody (D16), a time point of high inflammatory response due to the intravitreal injection. The second sample was taken close to the end of the experiment (D30). VEGF-A levels only slightly increased from D16 to D30 in the IgG control (from 52.8 ± 6.5 to 62.9 ± 27.7 pg/ml). Anti-VEGF-A (ranibizumab) and both treatments of RG7716 reduced aqueous VEGF-A levels compared to isotype control being more significant at the early time point rather than the late time point. Anti-ANG-2 administration did not affect VEGF-A levels (Fig 10A). Aqueous ANG-2 levels increased in the isotype control from D16 to D30 (6.1 ± 2.2 to 54.5 ± 29.7 pg/ml), while all other treatments reduced ANG-2 levels at D30 (Fig 10B). IL-6 levels slightly increased (from 89.6 ± 55.1 to 102.7 ± 78.6 pg/ml) in the aqueous of IgG control animals from D16 to D30. Reductions in IL-6 levels were achieved with all antibodies at D30, but no early reduction in IL-6 levels was evident 1 day after intravitreal injection (D16) (Fig 10C).

**Table 2. Positions of the exchanged amino acids are detailed below with the numbering according to Kabat and Wu (1991). In anti-VEGF-A/ANG-2-FcRγ⁻, the following substitution have been introduced: Leu234Ala, Leu235Ala, and Pro329Gly, and for anti-VEGF-A/ANG-2-FcRn⁻: Ile253Ala, His310Ala, and His435Ala.**

| | Short name | Wt | Position | Mutation | Affected Fc receptor |
|---|---|---|---|---|---|
| Anti-VEGF-A/ANG-2-FcRγ⁻ (Schlothauer *et al*, 2013) | PG | P | 329 | G | FcγRI, II, III |
| | LALA | LL | 234, 235 | AA | |
| Anti-VEGF-A/ANG-2-FcRn⁻ | AAA | I | 253 | A | FcRn |
| | | H | 310 | A | |
| | | H | 435 | A | |
| Anti-VEGF-A/ANG-2-FcRn-FcRγ⁻ | RG7716 | P | 329 | G | FcγRI, II, III, FcRn |
| | | LL | 234, 235 | AA | |
| | | I | 253 | A | |
| | | H | 310 | A | |
| | | H | 435 | A | |

ANG-2, angiopoietin-2; VEGF-A, vascular endothelial growth factor-A.

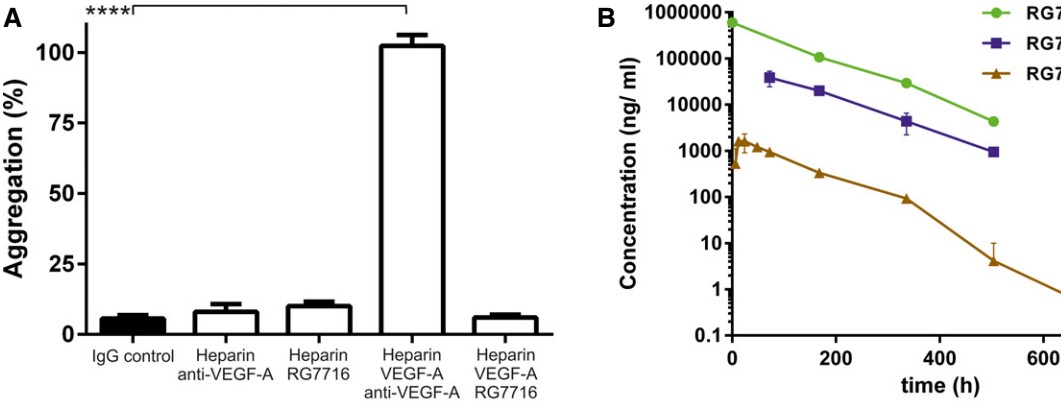

| C | | RG7716 | | Anti-VEGF-A/ANG-2-FcγR⁻ | |
|---|---|---|---|---|---|
| | | **Serum** | **Aqueous** | **Serum** | **Aqueous** |
| $C_{max}$ | μg/ml | 3.8 | 99 | 4.6 | 68.5 |
| $t_{max}$ | h | 24 | 72 | 24 | 72 |
| $t_{1/2}$ | h | 89.3 | 68 | 143 | 66 |
| $t_{last}$ | h | 672 | 672 | 588 | 672 |
| $AUC_{0-tlast}$ | (μg*h)/ml | 295 | 18100 | 622 | 13100 |
| $AUC_{0-inf}$ | (μg*h)/ml | 296 | 18200 | 655 | 13100 |
| F | % | 12.7 | NA | NA | 15.4 |

**Figure 8.  No platelet aggregation by RG7716 when antibody is complexed with VEGF-A and heparin *in vitro* and pharmacokinetic properties of RG7716 in comparison with wild-type anti-VEGF-A/ANG-2.**

A    Bar graph demonstrating platelet aggregation induced by complexes of heparin, VEGF-A$_{165}$, and anti-VEGF-A (IgG$_1$) *in vitro* using washed human platelets. Introduction of modification of the Fc region in RG7716 does not allow platelet aggregation to happen as compared to anti-VEGF-A as wild-type IgG$_1$. All three components are needed as heparin/anti-VEGF-A alone does not activate platelet aggregation. Error bars show SEM of three independent experiments and **** denotes significance of heparin/VEGF-A/anti-VEGF-A comparing to IgG control using ANOVA ($P < 0.0001$) and Dunnett's multiple *t*-test (****, $P < 0.0001$).

B    Kinetic presentation of mean serum, vitreous, and aqueous humor concentrations after single intravitreal administration of 0.5 mg of RG7716 in cynomolgus monkeys. Error bars indicate SEM with $n = 6$.

C    Summary table of key experimental PK parameters of RG7716 and anti-VEGF-A/ANG-2-FcγR⁻ (with intact FcRn binding).

Data information: AUC, area under the curve; $C_{max}$, maximum concentration; $t_{1/2}$, half-life; $t_{max}$, time to maximum serum concentration; $t_{last}$, time of collection of the last of a series of blood samples; F, bioavailability; NA, not analyzed; PK, pharmacokinetic.

RG7716 at 90 μg reduced the IL-6 levels the most and achieved significance. IL-8 levels in the aqueous were low, at an average of $3.2 \pm 1.4$ pg/ml on D16, rising to $16.4 \pm 8.7$ pg/ml on D30 in IgG control animals. All antibodies, except anti-ANG-2, reduced IL-8 levels to below 3 pg/ml at D30 (Fig 10D). Again, as noted for IL-6, RG7716 at 90 μg reduced IL-8 levels the most.

## Discussion

Diabetic retinopathy and wAMD are the leading causes of severe vision loss in middle-aged people and the elderly, respectively. Anti-VEGF therapies effectively reduce the growth of neovessels and the edema that is associated with these potentially blinding conditions, and thus have become the standard of care. However, despite the major medical advance, there is still an unmet patient need.

The efficacy of intravitreal ranibizumab has been demonstrated in patients with CNV secondary to wAMD in the pivotal phase III studies MARINA and ANCHOR (Rosenfeld *et al*, 2006a,b; Chang

*et al*, 2007; Brown *et al*, 2009). The most obvious pharmacodynamic effect seen with an anti-VEGF therapy is the reduction in retinal edema. Using this measure, patients can be divided into one of three categories: complete responders with no residual edema, incomplete responders with partial edema resolution, and nonresponders with no reduction in edema. Data from the pivotal trials suggest that 5–10% of the wAMD population are nonresponders and a large proportion of patients are incomplete responders. Data from the HARBOR trial (Ho *et al*, 2014) and aflibercept pivotal phase III trials VIEW 1 and VIEW 2 suggest that an efficacy ceiling has been reached with anti-VEGF monotherapy in the wAMD population, so novel therapeutic strategies are clearly needed. The next-generation therapies will need to shift the overall distribution of the patient response curve further to the responder side, and could potentially increase the proportion of patients who experience long-lasting improvements.

Using animal models of neovascular disease, we investigated the concept of dual vs. mono-inhibition with the aim to identify candidates that allow improving upon VEGF-A inhibitor monotherapies in the clinic. Here we describe preclinical studies focused on

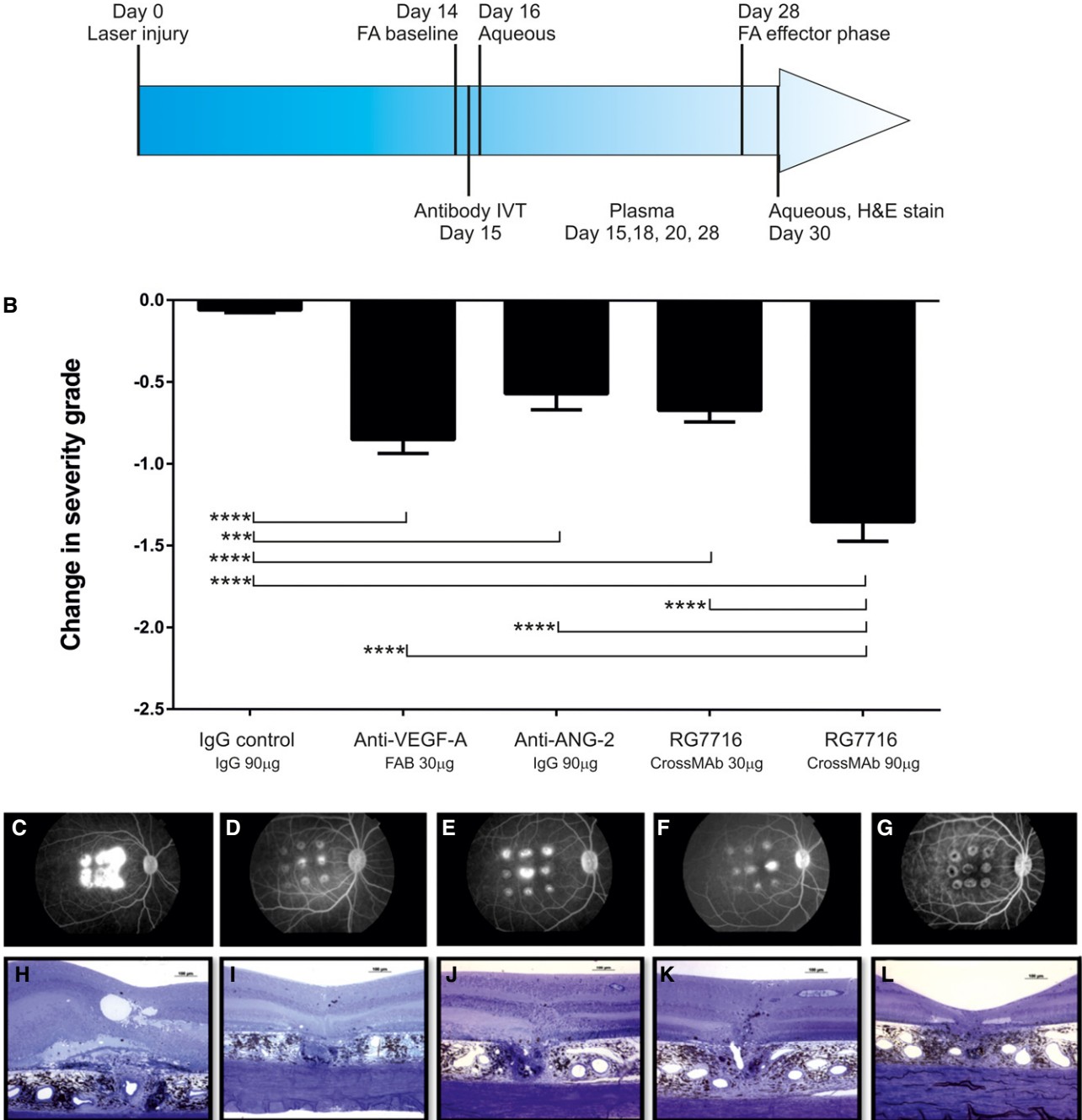

**Figure 9. Efficacy of RG7716 in a laser-induced CNV model using cynomolgus monkeys, comparing RG7716 to anti-VEGF-A (ranibizumab) and anti-ANG-2 (LC10), as well as an IgG control.**

A    Schematic representation of the experimental setup including treatment, sampling, and efficacy readouts.

B    Inhibition of neovascularization measured in severity grades, change of severity from baseline is shown for each treatment; all treatment significantly reduced the severity grade compared to IgG control. In addition, efficacy of RG7716 (150-kDa molecule at 90 μg/50 μl injected IVT) was significantly better at equal molar concentration of binding sites than anti-VEGF-A (ranibizumab, 50-kDa molecule at 30 μg/50 μl injected IVT), anti-ANG-2, and the low dose of RG7716 (30 μg/50 μl injected IVT). Error bars show SEM of $n = 6$ cynomolgus monkeys and nine spots per eye in the group; * denotes significance after one-sided ANOVA and Tukey's multiple $t$-test. IgG control is significantly different from anti-VEGF-A (****, $P < 0.0001$), anti-ANG-2 (***, $P = 0.0003$), RG7716, 30 μg (****, $P < 0.0001$), and RG7716, 90 μg (****, $P < 0.0001$). Furthermore, RG7716, 90 μg is significantly different from RG7716, 30 μg (****, $P < 0.0001$), anti-ANG-2 (***, $P = 0.0003$), and anti-VEGF-A (***, $P < 0.0004$).

C–L    (C–G) Representative figures of fluorescence fundus angiograms (H–L) of cross sections of a hematoxylin staining of a spot at the end of the experiment from eyes treated with IgG control, anti-VEGF-A, anti-ANG-2, and RG7716 at 30 and 90 μg, respectively.

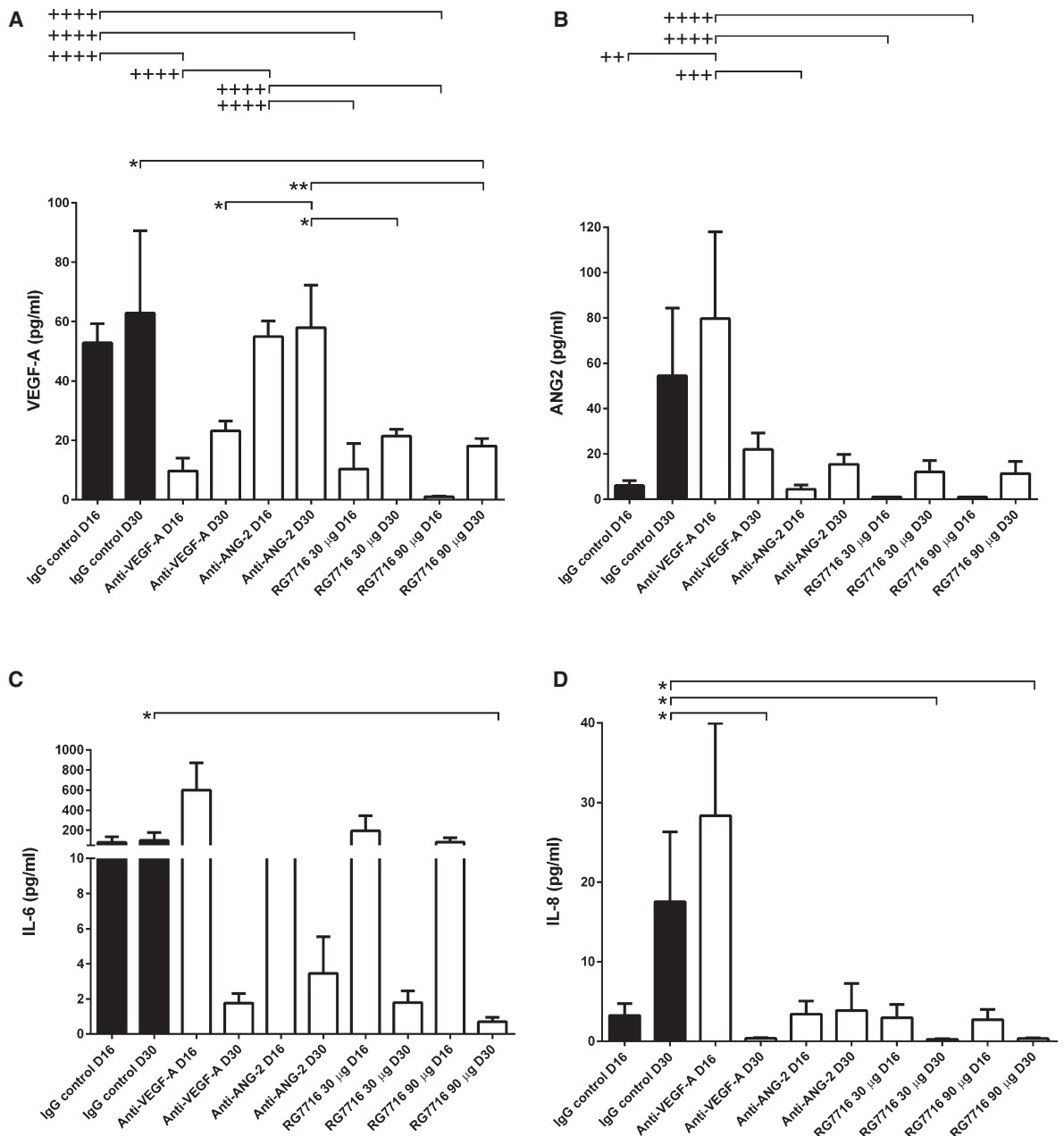

**Figure 10. Aqueous concentrations of VEGF-A, ANG-2, IL-6, and IL-8 samples 1 day after intravitreal antibody delivery (D16) and at the end of the experiment (D30) in non-human primate laser-induced CNV.**

A   VEGF-A levels only slightly increased from D16 to D30 in the IgG control. At D16, levels of IgG control are significantly higher compared to anti-VEGF-A ($^{++++}$, $P < 0.0001$); RG7716, 30 μg ($^{++++}$, $P < 0.0001$) and RG7716, 90 μg ($^{++++}$, $P < 0.0001$). Anti-ANG-2 levels are comparable to IgG control and therefore significantly different from anti-VEGF-A ($^{++++}$, $P < 0.0001$); RG7716, 30 μg ($^{++++}$, $P < 0.0001$) and RG7716, 90 μg ($^{++++}$, $P < 0.0001$). At D30, the levels of VEGF-A are significantly reduced in the RG7716, 90 μg group compared to IgG control (*, $P = 0.0309$). Anti-ANG-2 remains at levels comparable to IgG control and is therefore significantly higher than RG7716, 30 μg (*, $P = 0.0107$), RG7716, 90 μg (**, $P = 0.002$), and anti-VEGF-A (*, $P = 0.0162$).

B   ANG-2 levels increased from D16 to D30 in the IgG control. The same patterns, with a weaker total increase, were seen for the ANG-2 and both RG7716 treatment groups. ANG-2 levels were significantly higher at D16 in the anti-VEGF-A-treated group compared to IgG control ($^{++}$, $P = 0.002$), anti-ANG-2 ($^{+++}$, $P = 0.0004$), RG7716, 30 μg ($^{++++}$, $P < 0.0001$), and RG7716, 90 μg ($^{++++}$, $P < 0.0001$). The reductions of treatments compared to IgG control at day 30 did not reach significance.

C   IL-6 levels only very slightly increased from D16 to D30 in the IgG control, while for all other treatment groups there was a trend to reduced IL-6 levels at D30 compared to baseline with the high dose of RG7716 reaching significance, IgG control vs. RG7716, 90 μg (*, $P = 0.023$).

D   IL-8 levels increased from D16 to D30 in the IgG control group. Compared to the IgG control, the levels of IL-8 were reduced in the anti-VEGF-A (*, $P = 0.0262$), RG7716, 30 μg (*, $P = 0.0188$), and RG7716, 90 μg (*, $P = 0.0107$) treatments at D30.

Data information: Error bars show SEM, $^+$ and * denotes significance using one-sided ANOVA and Tukey's multiple $t$-test with $n = 6$ cynomolgus monkeys. Data are presented in linear scale; for the statistical analysis, data were transformed logarithmically to normalize distribution and allow parametric ANOVA testing. D16 (+) and D30 (*) were analyzed independently. IL-6, interleukin-6; IL-8, interleukin-8; D, day.

the dual targeting of ANG-2 and VEGF-A, key angiogenesis factors that are upregulated in many human neovascular conditions. Simultaneous inhibition of VEGF-A and ANG-2 was shown to be superior to monotherapy in the reduction in neovascular area in both early and late intervention studies in the mouse model of spontaneous CNV. Early intervention in this model was used to establish the principle that dual inhibition of VEGF-A and ANG-2 could reduce neovascularization and vessel leakiness. Follow-up experiments using mice in which antibodies were delivered when retinal lesions were already established demonstrated that combined inhibition and anti-ANG-2 alone strongly reduced lesion permeability. Late interference in the model did not reduce lesion numbers as strongly compared to early intervention, as neovascularization has taken place before antibody is given. Furthermore, the efficacy of anti-ANG-2 alone was as strong as the bispecific anti-VEGF-A/ANG-2 treatment at reducing lesion area in late interference. The results may highlight a more prominent role for VEGF-A mediating neovascularization and ANG-2 mediating vessel leakiness. At this stage, it is also not clear whether the small reduction in leaky lesion number in the late intervention model is due to reduction in new lesions generated at the late time point or by induction of lesion regression, a mechanism of action proposed for the VEGF-A/PDGF-B dual targeting (Jo et al, 2006; Nishijima et al, 2007). However, permeability was strongly reduced by the anti-VEGF-A/ANG-2 dual inhibition. The concept of strong control of vascular permeability by anti-VEGF-A/ANG-2 is supported by the OCT data, which showed a substantial reduction in retinal edema following late intervening VEGF-A and ANG-2 dual inhibition. While anti-ANG-2 reduced acute vessel leakiness as measured by FA, reduction in retinal edema was stronger with anti-VEGF-A/ANG-2 as measured by OCT. This observation warrants further investigation. The resolution of edema will further assist the preservation of normal retinal architecture, and is likely to lead to the findings of reduced outer retinal cell loss and increased visual function, as measured by ERG.

Experiments using an acute model of ocular inflammation, and the CNV models, suggested that the mode of action upon dual targeting of VEGF-A and ANG-2 may include an anti-inflammatory component. VEGF-A and ANG-2 are both capable of enhancing proinflammatory signals in endothelial cells. Intravital microscopy was used by Fiedler and colleagues to demonstrate normal TNF-α-induced leukocyte rolling in the vasculature of ANG-2-deficient mice, but that rolling cells did not firmly adhere to activated endothelium due to lower levels of adhesion molecules on the surface of endothelial cells (Fiedler et al, 2006). Cellular experiments showed that ANG-2 promotes adhesion by sensitizing endothelial cells toward TNF-α and modulating the TNF-α-induced expression of endothelial cell adhesion molecules. In addition, a recent study reported that a mild (two- to threefold) overexpression of VEGF-A in mice was sufficient to generate multiple pathological responses in the eye, including increased oxidative stress and activation of the NLRP3 inflammasome, cataract formation, and non-exudative AMD-like pathologies. IL-1 receptor-1-deficient mice did not show this phenotype, demonstrating the importance of pro-inflammatory mediators in enhancing signals that originate from VEGF-A (Marneros, 2016).

In our studies, we observed that the bispecific antibody was superior to either of the monotherapies in the reduction in

leukocyte infiltration in the LPS-induced inflammation model. Interestingly, ANG-2 inhibition alone was only mildly inhibitory, which was unexpected, since ANG-2 is often causally linked with inflammatory responses. LPS is known to rapidly trigger ANG-2 release and inhibition of ANG-2 is sufficient to reduce sepsis mortality in different rodent models (Ziegler et al, 2013; Menden et al, 2015). In addition, in the model of sCNV, we report increased numbers of IBA-1[+] macrophages around lesions and, in the laser CNV model, a reduction in aqueous levels of the cytokines IL-6 and IL-8 upon VEGF-A blockade or dual VEGF-A/ANG-2 inhibition; however, inhibition of ANG-2 alone had little effect. One likely explanation is that there are redundant pathways activated during major pathological insults, with the presumed biological aim of ensuring a robust and rapid response. If true, combination therapy approaches would certainly provide an advantage when responses become deleterious, and, in this current study, the anti-angiogenic, anti-edema, and anti-inflammatory advantages provided by dual targeting provide compelling reasons to further pursue this therapeutic strategy.

There are two primary drug-targeting strategies for testing whether the concept of dual inhibition of VEGF-A and the angiopoietin pathway translates to human disease: use of an anti-VEGF-A plus exposure to ANG-1 protein or inhibition of ANG-2 activity. Various approaches have been used to convincingly demonstrate that ANG-1 is important for maintaining vascular integrity and its activity is overall anti-angiogenic. The transgenic expression of ANG-1 resulted in leakage-resistant blood vessels (Thurston et al, 2000). Alternatively, adenoviral delivery (AAV2) of a recombinant version of ANG-1, called COMP-ANG-1, demonstrated a protective function of ANG-1 in models of choroidal neovascularization and diabetic retinopathy (Cahoon et al, 2015; Lambert et al, 2016). An oligomeric Tie-2 binding peptide called vasculotide showed protection against vascular leakage and mortality in a murine model of polymicrobial abdominal sepsis (Kumpers et al, 2011); however, the mode of action of this reagent is currently not fully understood (Wu et al, 2015).

While these approaches provided strong evidence for the protective role of ANG-1 for normalizing vessel function, they also have underscored, over the past two decades, the difficulty in producing a version of ANG-1 protein that is amenable to therapeutic approaches in the clinic.

Inhibition of ANG-2 was also shown in several preclinical oncology, sepsis, and ophthalmology models, to reduce pathological angiogenesis and vascular permeability (Saharinen et al, 2010; Campochiaro, 2013; Ziegler et al, 2013). The precise mechanism of action of ANG-2 appears to be complex, but can be partially explained by competitive antagonism of ANG-2 with ANG-1 binding to the Tie-2 receptor (Maisonpierre et al, 1997) and being an amplifier of apoptotic signals in pericytes under stress (Cai et al, 2008; Park et al, 2014).

Thus, we opted for inhibition of ANG-2 with classic therapeutic reagents—in this case, neutralizing antibodies. A bispecific CrossMAb antibody format offers the advantage of having to deliver only a single molecule with a unique set of molecular properties by intravitreal injection to neutralize two targets at once, in our case VEGF-A and ANG-2. Co-formulation of antibodies or biologics using the same paratope as in a CrossMAb is an alternative way to deliver therapeutic drugs. In cases where one ligand is in large excess of the

other, a co-formulation approach seems preferable to be able to increase the dose of the reagent neutralizing the ligand in excess. We consider the bispecific approach particularly suitable for soluble ligands, like VEGF-A and ANG-2, which are detected at roughly similar and low concentrations in fluids from the eye. VEGF-A levels are reported to be in a range of 51.8–454 pg/ml in RVO (Noma *et al*, 2010; Koss *et al*, 2011, 2014). Slightly higher levels (792 ± 203) are reported for macular edema in aqueous fluids (Funatsu *et al*, 2003). As reported here, these values for ANG-2 range from 139 pg/ml in wAMD to 1,625 pg/ml in pDR (Fig 1). Therefore, both ligands are available in comparative concentrations and both paratopes of the CrossMAb have an equal probability of neutralizing their respective ligands.

We designed RG7716 as a human $IgG_1$-like CrossMAb. The CrossMAb is a bispecific molecule design with one antigen-binding site binding VEGF-A and the second one binding ANG-2. Point mutations in the CH3 domain ("knobs-into-holes") promote the assembly of the two different heavy chains (Fig 6A). Exchange of the CH1 and CL domains in the ANG-2-binding Fab promotes the correct assembly of the two different light chains. This simple asymmetric molecule design enables large-scale pharmaceutical production in Chinese hamster ovary (CHO) cells (Schaefer *et al*, 2011). The advantage of this process lies in the ability to use a well-established IgG production work stream, including a standard CHO fermentation.

RG7716 is capable of binding, neutralizing, and depleting both human VEGF-A and ANG-2 simultaneously. Binding of the antibody to ANG-1 was not detected, nor did it have inhibitory activity in an ANG-1-driven endothelial cell survival assay using up to 100 μg/ml of RG7716. This property was deemed extremely important, since antibodies are delivered by intravitreal injection at high concentration (often milligrams/ml vitreous). Furthermore, while we detected strongly elevated levels of ANG-2 in vitreous fluids of patients, we also noted that ANG-1 is clearly present in the vitreous. Therefore, all necessary components of the ANG-1/ANG-2/Tie-2 system are present and selectivity is key to inhibition of ANG-2 to rebalance the ANG-1/ANG-2 ratio in favor of ANG-1 signaling for it to promote anti-angiogenesis, anti-vascular permeability, and vascular stability.

Since the strategy for achieving efficacy with RG7716 is based on ligand neutralization, we rendered the antibody Fc region without immune effector functions to optimize it for ophthalmological use. VEGF-A accumulates in the extracellular matrix and, in the presence of anti-VEGF-A, has been shown to activate platelets via the FcγRIIa receptor (Meyer *et al*, 2009) (Fig 8A). Activation of thrombocytes in the choroid was reported by Julien *et al* (2014), who compared ranibizumab and Fc-containing aflibercept 7 days after intravitreal injection. Significantly, higher amounts of free hemoglobin and protein complexes were found in the group treated with aflibercept, which contains an immune effector function-competent Fc fragment. It is anticipated that the Fc-engineered RG7716 has no similar potential for platelet aggregation.

The other important interaction mediated by the Fc part of an $IgG_1$ is the interaction with recycling receptor FcRn. The systemic half-life of $IgG_1$ is kept high due to the interaction with FcRn, which recycles $IgG_1$ by preventing the antibody from being degraded in the lysosome. We demonstrated that the systemic half-life of RG7716 is considerably shorter than that of a normal IgG in the cynomolgus monkey after intravitreal administration.

Importantly, the aqueous half-life of RG7716 was similar compared to the same antibody with the FcRn binding site intact (anti-VEGF-A/ANG-2-FcγR⁻ in Fig 8B and C). In the non-human primate CNV model, we also directly compared non-Fc region modified $IgG_1$s, a Fab fragment, and RG7716, and demonstrated comparable concentrations in aqueous fluid; however, RG7716 was cleared faster from the systemic circulation than a wild-type $IgG_1$ competent for binding to FcRn (Appendix Fig S4A and B). Anti-VEGF therapeutics have the potential to affect the systemic cardiovasculature; therefore, rapid systemic clearance is a desirable feature for such an ophthalmic drug delivered by intravitreal administration.

Our investigation also shows that the overall contribution of FnRn-mediated clearance of antibodies from the eye is small. Kim *et al* (2009) reported experiments that led the authors to conclude that FcRn plays an important role in eliminating intravitreally administered full-length IgGs across the blood–retinal barrier into the systemic circulation. One experiment used chicken IgY, which does not bind FcRn, and did not cross the blood–retinal barrier. Intravitreal injection of $hIgG_1$ (bevacizumab) resulted in less antibody crossing the blood–brain barrier in FcRn-deficient mice compared to wild-type IgG. These experiments would argue that elimination of FcRn binding may increase the vitreal half-life of RG7716; however, our results are not in line with previous work and show that this effect is small. Our experimental evidence suggests a mainly passive transport mechanism of RG7716 out of the eye, which would favor longer residence time for a larger molecule due to an increase in hydrodynamic radius.

We chose the cynomolgus monkey CNV model of laser injury to directly compare RG7716 to standard of care in clinical practice: the anti-VEGF-A ranibizumab. The weak interaction of ranibizumab with rodent VEGF-A did not allow comparisons in rodent models. One advantage of the cynomolgus model is that antibody delivery is by intravitreal injection, just as it is performed in the clinical setting. We also targeted a later-timed intervention, 15 days after the initial laser injury, to better reflect the clinical setting, where patients present with established lesions. RG7716 showed a significantly greater reduction in vessel permeability, as measured by fundus FA, compared with monotherapy using equimolar concentrations. However, a limitation of this laser model is that only low doses of anti-angiogenics can be investigated, since administration of as little as 30 μg of ranibizumab delivered by IVT were already efficacious. The potential additional benefit of RG7716 in these respects may finally only be demonstrable in clinical studies. Therefore, the current experiment aims to establish the scientific basis of additional benefit of RG7716 over ranibizumab monotherapy in the model of laser-induced CNV in cynomolgus monkeys using a subclinical dose delivered by intravitreal injection. These results support further dose-escalation studies in humans.

In conclusion, the future of ophthalmological advances is likely to be dependent upon the use of combination strategies to improve patient outcomes, similar to oncology. Dual inhibition of VEGF-A and ANG-2 showed increased efficacy compared to inhibition of VEGF-A or ANG-2 alone, in several relevant preclinical models of retinal neurovascular disease. RG7716 is a fully human bispecific antibody capable of binding to all isoforms of VEGF-A and, simultaneously, to ANG-2 without any binding to ANG-1 and features a modified Fc region, which results in lower

systemic concentrations compared to wild-type IgG$_1$ antibodies, and a reduced potential for platelet activation. These efficacy and pharmacokinetic data have been instrumental in supporting the testing of RG7716 in humans for its potential in ophthalmological indications. Appropriate safety studies have been conducted for progression into single- and multiple-dose phase I studies in patients with wAMD (NCT01941082). Currently, two phase II proof-of-concept studies are underway to probe for improved efficacy in humans in wAMD (NCT02484690) and DME (NCT02699450).

# Materials and Methods

## RG7716 antibody production and Fc mutations

### Cloning
The fusion genes comprising the antibody chains as described below were generated by polymerase chain reaction (PCR) and/or gene synthesis and assembled by known recombinant methods and techniques. To disable FcγR and FcRn binding, the mutations were introduced in the human IgG framework as described in Tables 1 and 2 (T. Schlothauer *et al*, unpublished data). For the expression of the described antibodies, expression plasmids for transient expression in HEK293-F cells (Invitrogen Corporation) were generated.

### Expression
Bispecific antibodies were generated by transient transfection with the four respective plasmids (encoding the heavy and the crossed heavy chain, as well as the corresponding light and crossed light chain) using the HEK293-F system (Invitrogen Corporation) according to the manufacturer's instructions.

### Purification
Bispecific antibodies were purified from cell culture supernatants by affinity chromatography using MabSelectSure-Sepharose (for non-FcRn binding mutants) (GE Healthcare Europe GmbH) or kappa Select-Agarose (for FcRn binding mutants) (GE Healthcare), hydrophobic interaction chromatography using butyl-Sepharose (GE Healthcare), and Superdex 200 size-exclusion (GE Healthcare) chromatography.

The bispecific antibody-containing fractions were pooled, concentrated to the required concentration using Vivaspin ultrafiltration devices (Sartorius Stedim Biotech S.A.), and stored at −80°C in 20 mM histidine, 140 mM NaCl, pH 6.0.

### Analytics
Purity and antibody integrity were analyzed after each purification step by capillary electrophoresis–sodium dodecyl sulfate (CE-SDS) using microfluidic Labchip technology (Caliper Life Science). The aggregate content of antibody samples was analyzed by high-performance size-exclusion chromatography (SEC) using a Superdex 200 analytical size-exclusion column (GE Healthcare) in 2× PBS running buffer at 25°C.

### Surrogates
To evaluate the efficacy of a combined anti-VEGF-A and anti-ANG-2 treatment in mouse models, we used B20-4.1 as a surrogate to

inhibit VEGF-A in the appropriate molecular format (Fab, IgG, or CrossMAb) (Liang *et al*, 2006).

All FcRn, FcγR, and simultaneous binding experiments were measured on a BIAcore (GE Healthcare) system.

### FcRn binding
Overlays were measured as described by Schlothauer *et al* (2013).

### FcγR binding overlay
Around 6,000 RU of the capturing system (1 μg/ml Penta-His antibody; Qiagen) were coupled on a CM5 chip (GE Healthcare Europe GmbH) at pH 5.0 using an amine coupling kit. The sample and system buffer was HEPES-buffered saline, pH 7.4. FcγRI/IIa/IIIa-His were captured by injecting a 100 nM solution for 60 seconds at a flow of 5 μl/min. Binding was measured by injection of 100 nM of CrossMAb for 180 s at a flow of 30 μl/min.

### Assessment of independent VEGF-A and ANG-2 binding to the CrossMAb
Around 3,500 RU of the capturing system (10 μg/ml goat anti human IgG) were coupled on a CM4 chip at pH 5.0 by using an amine coupling kit (all reagents and buffers by GE Healthcare). The sample and system buffer was PBS-T (10 mM PBS including 0.05% Tween-20) pH 7.4. The bispecific antibody was captured by injecting a 10 nM solution for 60 s at a flow of 5 μl/min. Independent binding of each ligand to the bispecific antibody was assessed by order-of-addition experiments (human VEGF-A concentration 200 nM, human ANG-2 concentration 100 nM). The surface was regenerated by 60 sec washing with a 3 mM MgCl$_2$ solution at a flow rate of 30 μl/min.

### Isothermal titration calorimetry
The affinities of the human (h)ANG-2 RBD-Fc fusion proteins, hVEGF-A$_{121}$ and hVEGF-A$_{165}$, for RG7716 were measured using the isothermal titration calorimeter iTC200 (GE Healthcare). hANG-2-RBD-Fc, hANG-1-RBD-Fc, and RG7716 were all dialyzed against 15 mM HEPES, 200 mM NaCl, pH 7.4. Ligands were loaded into the sample cell at a concentration of 4 μM and RG7716 applied via syringe at a concentration of 60 μM by 12 injections of 3 μl following a single injection of 1 μl to equilibrate the system. VEGF-A$_{121}$, VEGF-A$_{165}$, and RG7716 were dialyzed against 1× PBS. VEGF-A$_{121}$ or VEGF-A$_{165}$ was loaded into the sample cell at concentrations of 4 μM or 2 μM, respectively. The antibody solution was applied via syringe at a concentration of 40 μM via 17 injections of 2 μl following a single injection of 0.5 μl to equilibrate the system. The experimental titration curves were fitted to a one-site binding model using Origin®7 (OriginLab Corporation).

### ANG-1 selectivity and TEER assay using endothelial cells
Stem cell-derived endothelial cells were grown on fibronectin-coated flasks up to passage 3 and resuspended in assay medium (EBM-2 + 0.5% FCS) and transferred to fibronectin-coated 96-well plates using 12,000 cells/well (Patsch *et al*, 2015). After 24-h incubation, 120 ng/ml ANG-1 was added together with increasing concentrations of antibody (up to 100 μg/ml) and incubated for 3 days. Cell viability was measured with alamarBlue® after 3-h incubation measuring absorbance of alamarBlue® at 570 nm, using 600 nm as a reference wavelength.

For TEER measurements, fibronectin-coated 24-well transwell inserts were seeded with $1 \times 10^5$ primary human endothelial cells in

100 µl growth medium (EGM™-2). After 2 days, the medium was changed to endothelial assay medium (EBM™-2 supplemented with SingleQuots® kit excluded VEGF-A, only with 0.5% FCS) and cultured for 2 more days before the transwells were placed into the CellZscope® (NanoAnalytics). TEER was then measured continuously in a $CO_2$ incubator at 37°C. After 8 h, cells were treated with VEGF-A (basal) and ANG-1 or anti-ANG-2 (apical and basal), respectively, and continued for TEER measurements. ANG-2 levels of primary endothelial supernatants were measured with the DUO-Set human ANG-2 ELISA from R&D according to the instruction leaflet. Supernatants have been diluted in reagent diluent from R&D and TMB substrate used for detection.

### Washed platelets: preparation and aggregation

Human blood was collected in 85 mM trisodium citrate, 67 mM citric acid, glucose 111.5 mM, pH 4.5, in final concentration 0.02 U/ml apyrase and 0.5 µM prostacyclin (PGI2) was added as an anticoagulant. Platelet-rich plasma (PRP) was prepared and platelets were resuspended in 20 mM NaCl, 13 mM trisodium citrate, 30 mM dextrose, pH 7.0 + 1 µM PGI2 + 0.02 U/ml apyrase before being twice pelleted (760 *g* at room temperature) and was resuspended in buffer as before. After the third centrifugation, the pellet was resuspended in ETS buffer (37°C) (154 mM NaCl, 10 mM Tris–HCl, 1 mM EDTA, pH 7.4) + 1 µM PGI2 + 0.02 U/ml apyrase. After a last centrifugation (760 *g* at room temperature), the pellet was finally resuspended in Tyrode's buffer (37°C) (133 mM NaCl, 2.7 mM KCl, 11.9 mM $NaHCO_3$, $NaH_2PO_4$ 0.36 mM, 10 mM HEPES, 5 mM glucose, 2 mM $CaCl_2$, 1 mM $MgCl_2$, BSA 0.2%; pH 7.4) + 0.02 U/ml apyrase. After resting for an hour at 37°C, platelet concentration was adjusted to $2 \times 10^8$ platelets/ml. A total of 300 µl of washed platelets was added to each well of a 32-channel aggregation plate (custom product, equivalent to a 32-channel aggregometer), stirred for 3 min, and primed in the presence of 2.5–5 mM of adenosine diphosphate. After 20 s, 20 µl of antibody/VEGF-A/heparin mixtures was added and, another 3–5 min later, 20 µl of fibrinogen was added. Readout was percentage aggregation.

### Measurement of ANG-1 and ANG-2 levels in vitreous fluids from patients with retinal disease

Vitreous samples of patients with wAMD, diabetic retinopathy, retinal vein occlusion, and macular hole were collected by Dr. M. Koss, University of Heidelberg (Koss *et al*, 2011). Collection of samples was performed after local full ethics committee approval (57/08) in accordance with the European Guidelines for Good Clinical Practice and the Declaration of Helsinki. Informed consent was obtained from each patient before the start of therapy. Samples were stored at −70°C and transferred to Roche for the analysis of ANG-1 and ANG-2 using a multiplex Luminex kit produced in-house using the following reagents: anti-ANG-1 capture (R&D, MAB9231) and anti-ANG-1 detection (Novus Biologicals, NB110-85464), anti-ANG-2 capture (R&D, MAB098) and anti-ANG-2 detection (R&D, BAM0981). Capture antibodies were coupled to Luminex beads using the Bio-Plex Amine Coupling Kit (Bio-Rad 171406001). Vitreous samples diluted one-third were incubated with capture antibody bead for 2 h. After washing the beads, biotinylated detector antibodies were added and incubated with the beads for

1 h. Streptavidin-conjugated fluorescent protein, R-phycoerythrin (BD, 554061), was then added and incubated for 30 min. After washing, the beads were analyzed using a Luminex 100 detection system.

### Spontaneous CNV model in JR5558 mice

All procedures on mice were ethically reviewed and approved according to the British Home Office Animals Scientific Procedures Act 1986 and were performed in accordance with European Directive 86/609/EEC, the Roche Ethics Committee on Animal Welfare (ECAW), and the statement for the use of animals in ophthalmic and vision research approved by the Association for Research in Vision. Fundus FA was performed on unanesthetized mice using a Digital Kowa Genesis Fundus camera for small animals (Kowa OptiMed, Tokyo, Japan). Imaging was initiated once the pupils were fully dilated (~2 mm diameter). Sodium fluorescein (NaF) (2%) was injected IP (13.5 µl/g body weight). Early-phase angiograms were captured 90 s after injection, followed by late-phase angiograms at 5 min. Multiple images were captured at each time point covering the central retinas (~75° field of view). Areas of fluorescein leakage adjacent to angiogenic lesions were manually selected and number of, area, and brightness in early- and late-phase angiographies measured (ImageJ, Wayne S. Rasband, NIH, Bethesda, Maryland, USA).

Optical coherence tomography was performed using the Heidelberg Spectralis with a +25 diopter achromatic lens attached to the front of the device. A custom-made PMMA contact lens (Cantor & Nissel Ltd, UK) was used to avoid dehydration of the cornea of anaesthetized mice. OCT scans were obtained from a 1.5-mm$^2$ region at 10-µm intervals of the central nasal retina simultaneously with FA (see above). The data were analyzed using a Canny edge-based segmentation algorithm, which delineated the outer segment and retinal pigment epithelium, and the volume separating the two was calculated (MATLAB). Data sets collected from the same region and eye of the same animal at baseline, after 1 week, and at the end of the study were geometrically aligned and change in subretinal volume quantified.

Standard full-field scotopic and photopic large-field ERGs were recorded from dark-adapted (12 h) and light-adapted mice (Micron III with ERG adapter, Phoenix Research Laboratories). Scotopic recordings were performed under dim red light. Series of 5-ms single-flash recordings were obtained at increasing light intensities from −2.5 to 3.0 log cds/m$^2$. Twenty responses per intensity were averaged with an interstimulus interval of 20 s. A- and B-wave amplitudes and implicit times were evaluated (Labscribe, iWork System Inc.).

Ocular tissues were collected 24 h after the last FA. Eyes were removed and snap-frozen in isopentane on dry ice. From frozen eyes, vitreous, retina, and retinal pigment epithelium (RPE)/choroid samples were isolated. Alternatively, eyes were enucleated and fixed overnight in 4% paraformaldehyde for immunohistochemistry of whole-mounts of retinae and of the RPE/choroid complex (eyecup). Alternatively, eyeballs were cryoprotected in 30% sucrose and snap-frozen in Optimal Cutting Compound 4585 (OCT, Tissue-Tek; Miles, Elkhart, IN), and cryostat sections (10 µm) were thaw-mounted onto charged slides. Retinal sections were blocked for 1 h in 5% normal donkey serum in 0.1 M PBS, pH 7.4 with 0.3% Triton

X-100 and incubated for 2 h at room temperature with primary antibodies. After washing, slides were exposed for 1 h to fluorescently conjugated appropriate secondary antibodies. Nuclei were subsequently stained with 4′,6-diamidino-2-phenylindole (DAPI). Slides were washed for 3 × 5 min in 0.1 M PBS and finally cover-slipped using Vectashield (Vector Laboratories). TUNEL analysis was performed as recommended by the supplier, Promega DeadEnd Fluorometric TUNEL System. TUNEL-positive cells were counted in whole-mount preparations of neural retinas. Whole-mount retinas and eyecups were imaged using a laser scanning confocal microscope (Zeiss LSM 710; Carl Zeiss) at 8 bit/channel and 1,024 × 1,024 pixels. Sections were viewed and images captured on an epifluorescence bright-field microscope (Olympus BX50F4, Olympus), where data were captured as 36-bit color images at 3,840 × 3,072 pixel resolution using a Retiga SRV camera (QImaging).

## Single-dose pharmacokinetic study, intravitreal dose administration

A single-dose pharmacokinetic study was conducted by Covance Laboratories Inc. (Madison, Wisconsin, USA). All procedures in the study were carried out in compliance with the Animal Welfare Act Regulations (9 CFR 3). Six male cynomolgus monkeys received a single-dose intravitreal injection (50 μl with 1.5 mg/eye of RG7716). Blood was collected into serum separator tubes at indicated times post-dose. Serial aqueous humor samples were collected at 72, 168, and 336 h post-dose. Serum, aqueous, and vitreous humor samples were analyzed for RG7716 concentration with an ELISA that uses anti-idiotypic antibodies (Stubenrauch *et al*, 2013).

## Laser-induced CNV in cynomolgus monkeys

Experimentation on non-human primates (*Macaca fascicularis*) was performed in accordance with the Statement for the Use of Animals in Ophthalmic and Vision Research approved by the Association for Research in Vision and Ophthalmology. The guidelines of the Animal Ethics Committee of the Singhealth Singapore Association for Assessment and Accreditation of Laboratory Animal Care were also satisfied. In this study, five groups of six female cynomolgus monkeys at 3–5 kg body weight (*n* = 30) were used. CNV was induced by laser photocoagulation on D0 in both eyes of 30 cynomolgus monkeys using a 532-nm laser (PurePoint 532 nm Green Laser; Alcon) attached to a slit-lamp delivery system and a hand-held contact lens. Nine lesions were symmetrically placed in the macula of each eye by a masked retinal specialist. The parameters used were spot size (50 μm), duration (0.1 s), and 500 mW–1 W. The distance from each laser spot to the central fovea was maintained at 0.5–1 disk diameter size. Intravitreal injections of 50 μl antibody solution per eye were performed on both eyes on D15. Change in CNV grade was determined by comparing FA on D28 vs. D14. FA was performed using a commercial camera and imaging system (TRC-50DX Fundus camera; Topcon) at 14 and 28 days after laser photocoagulation. Photographs were captured with the fundus camera lens after intravitreal injection of 0.1 ml/kg of body weight of 10% fluorescein sodium. A masked retina specialist not involved in laser

photocoagulation or angiography evaluated the FA at a single sitting.

On D30, the animals were sacrificed and the upper body was perfused with half-strength Karnovsky's fixative. The eyes were removed, postfixed for 2–3 days in half-strength Karnovsky's fixative. Strips of tissue containing one or two lesion sites were embedded in plastic. Sections 2-μm thick were taken at 30-μm steps through the middle of each lesion and stained with toluidine blue.

Aqueous humor was collected from all eyes on D16 (1 day after intravitreal injection of antibody) and D30. Cytokine analysis of aqueous was performed by multiplex ELISA system measuring IL-6, IL-8, VEGF-A, and ANG-2. Blood was taken from a femoral or saphenous vein at indicated days. All samples were maintained at room temperature and allowed to clot, then chilled and centrifuged within 1 h of blood collection, and stored at −80°C.

**Expanded View** for this article is available online.

## The paper explained

### Problem

Age-related macular degeneration (AMD) is a major cause of vision loss in the elderly population. Severe vision loss is seen in the wet form of the disease. Critical initiator of disease are neovessels that grow from choroidal vessels to underneath and into the retina. These are immature in phenotype and are therefore leaky, leading to distortion of the tissue architecture, which impacts the functionality of the retina. Neutralization of VEGF-A, a major driver of vessel neoangiogenesis and leakiness, by biological reagents is currently the standard of care to treat wet AMD. However, medical need remains as not all patients gain visual acuity. It is therefore import to enhance the efficacy of VEGF-A-neutralizing reagents.

### Results

Angiopoietin-2 (ANG-2) is another important growth factor involved in neoangiogenesis and vessel leakiness. We tested whether dual inhibition of VEGF-A and ANG-2 is more efficacious compared to inhibition of VEGF-A alone in animal models of choroidal neovascularization. In a mouse model of spontaneous choroidal neovascularization, we saw dual inhibition reduces vessel leakiness and the total number of lesions. As a consequence, retinal functionality was protected as revealed by electroretinogram and the number of apoptotic cells in the retina was reduced. In a non-human primate model of laser-induced choroidal neovascularization, we confirmed that dual inhibition of VEGF-A and ANG-2 is more efficacious in reducing the leakiness of lesions after intravitreal delivery of neutralizing antibodies. For the non-human primate model, we used RG7716, which is a CrossMAb specifically designed for the application in ophthalmology. This antibody potently binds and neutralized VEGF-A with one antigen-binding arm and very selective anti-ANG-2 on the other. Its Fc domain is disabled for interaction with FcγR and FcRn reducing the potential for platelet activation and reducing the systemic half-life of the antibody.

### Impact

This study demonstrated that dual inhibition of VEGF-A and ANG-2 is more efficacious in preclinical models of choroidal neovascularization compared to inhibition of VEGF-A alone. We therefore generated a human CrossMAb RG7716, which has been tested in single- and multiple-dose phase I studies in patients with wet AMD (NCT01941082), and two phase II studies to test the concept of improved efficacy in proof of concept studies in humans are ongoing in wAMD (NCT02484690) and DME (NCT02699450).

## Acknowledgements

The authors thank Harald Duerr, Hubert Kettenberger, Ingo Gorr, Michael Molhoj, Sabine Imhof-Jung, Thomas V. Hirschheydt, Ulrich Goepfert, Dhananjay Jere, Wolfgang Schäfer, Matthias Rueth, Ulrike Reiff, and Markus Thomas for providing and testing of material used in this publication and for continuous support of this project; Yin-Shan Ng, Shannon Conder, and Joanna Holeniewska for assistance with the *in vivo* experiments in the sCNV model; and Greogor William Jainta for excellent technical support for the endothelial cell experiments. Roche provided funding sources for these studies to DTS and VAB.

## Author contributions

JTR and CK are co-initiators of the project, designed, constructed, and produced RG7716. JM was responsible for *in vitro* characterization. PLvL and DTS tested and analyzed in JR5888 mice. VAB, CMGC, SBBT, and YSW tested and analyzed the NHP model. MD analyzed PK experiments. KGS ran IgG ELISA in NHP samples. DI performed the LPS challenge model. MJK provided and EN analyzed clinical samples. RF provided ANG-2 mRNA analysis of JR mice, KGS and GW were responsible for cytokine ELISA and platelet aggregation assay. PS performed the endothelial cell experiments. DTS was senior advisor to the project. GH was project co-initiator, coordinator, and senior author.

## Conflict of interest

MD was an employee of Roche at the time of experimentation and is now an employee of Shire Pharmaceuticals. JTR, JM, and KGS are employees of Roche Diagnostics GmbH. GW, EN, CK, PS, and GH are employees of F. Hoffmann La Roche.

## For more information

AVENUE: A Proof-of-Concept Study of RG7716 in Participants With Choroidal Neovascularization (CNV) Secondary to Age-Related Macular Degeneration (AMD) https://clinicaltrials.gov/ct2/show/NCT02484690.

BOULEVARD: Phase 2 Study of RO6867461 in Participants With Center-Involving Diabetic Macular Edema (CI-DME) https://clinicaltrials.gov/ct2/show/NCT02699450.

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
