## [Review Process File · EMBO Molecular Medicine]

Targeting key angiogenic pathways with a bispecific CrossMAb optimized for neovascular eye diseases

Jörg T. Regula, Peter Lundh von Leithner, Richard Foxton, Veluchamy A. Barathi, Chui Ming Gemmy Cheung, Sai Bo Bo Tun, Yeo Sia Wey, Daiju Iwata, Miroslav Dostalek, Jörg Moelleken, Kay G. Stubenrauch, Everson Nogoceke, Gabriella Widmer, Pamela Strassburger, Michael J. Koss, Christian Klein, David T. Shima, and Guido Hartmann

Corresponding author: Guido Hartmann, F. Hoffmann-La Roche Ltd

Review timeline:	Submission date:	07 October 2015
	Editorial Decision:	17 November 2015
	Revision received:	30 May 2016
	Editorial Decision:	22 June 2016
	Revision received:	20 July 2016
	Editorial Decision:	09 August 2016
	Revision received:	21 August 2016
	Accepted:	02 September 2016

Transaction Report:

Editor: Roberto Buccione

1st Editorial Decision

17 November 2015

Thank you for the submission of your manuscript to EMBO Molecular Medicine. We have now heard back from three Reviewers whom we asked to evaluate your manuscript.

We are sorry that it has taken so long to get back to you on your manuscript. In fact, we experienced some difficulties in securing three willing and appropriate reviewers and in obtaining their evaluations in a timely manner and finally, there were persisting difficulties in retrieving a third. I will be thus making a decisions based on two consistent evaluations from very expert reviewers.

You will see that although the Reviewers find your work of potential interest, they point to overlapping concerns that impinge on the overall impact and robustness of the main conclusions. Although I will not dwell into much detail, I would like to highlight the main points.

In essence, the reviewers point to statistical issues, identify experimental design flaws including lack of fundamental controls (notably missing comparative analysis vs. both mono specific antibodies separately), unclear demonstration of actual superior efficacy of the bi-specific antibody as opposed for instance to sequential inhibition of VEGFA and ANG2, inconsistent antibody dosages, and

others.

You will also see that more mechanistic insight is requested. I am willing to forego this, provided the other issues are fully addressed, although I do encourage you to develop your study as far as realistically possible in a mechanistic sense for your next, revised version to strengthen your findings and increase their impact.

I would also understand if you are not able to carry out further experimentation on the NHPs. Should this be the case, please explain this in your point by point rebuttal. Provided the other issues are solved, this specific point will not be a basis for rejection.

In conclusion, while publication of the paper cannot be considered at this stage, given the potential interest of your findings, we would be pleased to consider a revised submission, with the understanding that the Reviewers' concerns must be addressed with additional experimental data where appropriate and that acceptance of the manuscript will entail a second round of review.

Please note that it is EMBO Molecular Medicine policy to allow a single round of revision only and that, therefore, acceptance or rejection of the manuscript will depend on the completeness of your responses included in the next, final version of the manuscript.

As you know, EMBO Molecular Medicine has a "scooping protection" policy, whereby similar findings that are published by others during review or revision are not a criterion for rejection. However, I do ask you to get in touch with us after three months if you have not completed your revision, to update us on the status. Please also contact us as soon as possible if similar work is published elsewhere.

Please note that EMBO Molecular Medicine now requires a complete author checklist (<http://embomolmed.embopress.org/authorguide#editorial3>) to be submitted with all revised manuscripts. Provision of the author checklist is mandatory at revision stage; The checklist is designed to enhance and standardize reporting of key information in research papers and to support reanalysis and repetition of experiments by the community. The list covers key information for figure panels and captions and focuses on statistics, the reporting of reagents, animal models and human subject-derived data, as well as guidance to optimise data accessibility.

Last, but not least, please carefully conform to our author guidelines (<http://embomolmed.embopress.org/authorguide>) to ensure rapid pre-acceptance processing in case of a favourable outcome on your revision.

I look forward to seeing a revised form of your manuscript in due time.

***** Reviewer's comments *****

Referee #1 (Remarks):

Regula et al have examined the effect of the bispecific VEGF and Ang2 targeting CrossMAb for ocular diseases in rodent and nonhuman primate models. The authors intended to demonstrate that targeting VEGFA and ANG2 at the same time is more effective than monotherapy. Overall, the study is a solid piece of work. However, its impact is somewhat limited because of a less than perfect experimental design, a limited scope and novelty of the results and the study's overall limited mechanistic impact.

First, it is not clear to the reviewer if the authors have fully considered the concepts of bioequivalence. The crossmab appears to be of similar MW and behavior as the parent IgG. Yet, has this been fully validated in terms of antibody efficacy and pharmacokinetics? Apparently, the authors use equal molar concentrations in some experiments and equal weight in others. Likewise, the employed concentrations appear to vary quite significantly from experiment to experiment (see below).

Second, it should be standard to compare the bispecific antibody with each single targeting antibody alone. However, the proper controls are missing in many experiments, most importantly the anti-ANG2 alone treatment group (see below).

Third, the authors only mention statistical analysis in the "laser-induced CNV in Cynomolgus monkeys" experiment, leaving out the statistical analysis in other experiments. And some differences, which may be statistically significant, are not marked as such (see below).

Beyond these technical limitations, the rationale of targeting VEGFA and ANG2 simultaneously is not solidly supported by the results. Fig 1G shows that anti-VEGFA/ANG2 did not have a superior effect in reducing lesion numbers compared to anti-VEGFA or anti-ANG2, respectively. However, anti-VEGFA/ANG2 and ANG2 were better at reducing lesion size as shown in Fig 1H. This argues that it may possibly be a better strategy to target VEGFA and ANG2 in a sequential manner.

Last but not least, the manuscript is not written in a well-organized and concise manner. The results should be re-organized into more specific sections with clearer subtitles. The discussion is too redundant. The authors should discuss interpretations of the results, not summarizing related findings. In short, better evidence will need to be provided, proper controls need to be included and the manuscript should be re-written in order to make it a solid contribution.

Beyond these general comments, the authors should consider the following specific suggestions:

1. The authors have examined different aspects of anti-VEGF-A/ANG-2 in the spontaneous CNV mouse model, which demonstrated promising efficacy. However, the experiments do not appear to be designed consistently.

1a) The dosages used in different experiments appear to be somewhat inconsistent. The authors administrated 5 mg/kg of antibodies in early intervention model and 3 mg/kg in the late-intervention model without giving reasons for the change of the dosage. To check the edema and the inflammation in the late interference model, the author then went for a range of concentrations of anti-VEGF-A/ANG-2, again without considering the previous dosage of 3 mg/kg.

1b) Proper controls are missing in some of the experiments. It should be consistent throughout the manuscript to compare the bispecific antibody with each of the single targeting reagents. However, the anti-ANG-2 treatment was left out in the early intervention model (Fig 1E and 1F) and macrophage infiltration staining (Fig 3A, 3C-E). The author mentioned in the early intervention model vehicle control, which could not be found in the figures and is also not included in other experiments.

1c) Some representative images are missing: in Fig 2A, anti-ANG-2 and anti-VEGF-A/ANG-2 at mid and high dosage; in Fig 3C-H, anti-ANG-2 and anti-VEGF-A/ANG-2 at low and high dosage; in Fig 3I, anti-VEGF-A and anti-ANG-2.

1d) The authors did not state clearly in the figure legends what the error bars refer to (SD or SEM). For some apparently obvious differences, there is no significance level marked. Fig 1H: there was a significant downregulation of anti-VEGF-A treatment but not anti-VEGF-A/ANG-2, which was more effective and with smaller error bar. Fig 2A demonstrated that low dosage of anti-VEGF-A/ANG-2 already had an obvious effect, whereas in Fig 2B there is no significance detected in the

low and middle dosage of anti-VEGF-A/ANG-2 treatment.

2. The rationale of targeting VEGF-A and ANG-2 simultaneously is not substantially supported by the results in the late-intervention spontaneous CNV mouse model. The authors show that anti-VEGF-A/ANG-2 was not better than either anti-VEGF-A or anti-ANG-2 in reducing lesion numbers and not better than anti-ANG-2 in reducing lesion area. On thus wonders, if it would not be better to target VEGF-A and ANG-2 in a sequential manner?

To demonstrate the superiority of the bispecific antibody and clearly lay out a good rationale, the authors definitely need to check the temporal expression of local VEGF-A and ANG-2 during the development of the disease and compare all the different targeting strategies in this aspect.

3. The authors generated the RG7716 with ANG-2i-LC10 binding domain. However, it was the ANG-2i-LC06 binding domain that was characterized in the cited publication (Schaefer et al, 2011). Why did the authors use the ANG-2i-LC10 instead of ANG-2i-LC06 here?

4. The authors addressed the question how the antibodies affect inflammation around the lesions by examining IBA-1-positive macrophages. IBA-1 is not a macrophage-specific marker. Therefore, the authors are asked to use another marker to demonstrate the results, for example CD68 or F4/80. In addition, as the authors have discussed, Tie2-positive macrophages can promote neoangiogenesis and more recently Ang2 has also been shown to recruit CCR2 positive macrophages via CCL2 and affect inflammatory pathways like NF-kb and Stat3 signaling. Hence, it would be interesting to further demonstrate which subpopulations of macrophages were affected by antibody treatment and provide more mechanistic insights. Isolectin is also not an endothelial specific marker. It can also be taken up by macrophages, making it unsuitable for such an analysis. Hence, another marker such as CD31 should likely be used instead. In addition, the authors claim in the text to stain endothelial cells with isolectin-B4. However, they wrote endomucin staining in the figure legend. Which marker did the authors actually use? The authors should also provide representative images of all treatment groups.

5. In the pharmacokinetic study of RG7716 in Cynomolgus monkeys, only concentrations for RG7716 were measured in vitreal humor, without proper controls. The same experiment including anti-VEGF-A/ANG-2-Fc R as comparison is therefore required.

6. The authors have done substantial work to examine the bispecific antibody which shows promising effects. Could the author provide some more mechanism-based insights to explain how the antibody achieves such an effect?

Referee #2 (Remarks):

Regula et al., Targeting key angiogenic pathways with a bispecific CrossMAb optimized for neovascular eye diseases

In their manuscript the authors present data on CrossMAb RG7716, a bispecific domain exchanged monoclonal antibody capable of binding, neutralizing, and depleting both VEGF-A and angiopoietin-2 (ANG-2). The authors use the spontaneous choroidal neovascularization (CNV) model in JR5558 mice and a nonhuman primate model of laser-induced CNV. The authors show that the simultaneous inhibition of VEGF-A and ANG-2 using RG7716 is effective in reducing vessel leakiness, CNV lesion number and in improving visual functionality - in some experiments more effectively than anti-VEGF or anti-Ang2 monotherapy. The authors also demonstrate modifications of the FcRn and Fc R binding sites of RG7716, which prevent Fc-mediated effector

functions of RG7716, resulting in increased systemic, but not ocular, clearance.

The manuscript deals with an important and interesting topic. There is an unmet need for better treatment options in CNV-associated diseases, such as age-related macular degeneration (AMD), since not all patients respond to current VEGF blocking therapies. In addition, previously published work has demonstrated that ANG-2 is an important contributor in CNV. However, based on the work by Regula et al., the real benefit of RG7716 in comparison to anti-VEGF or anti-Ang2 monotherapies remains elusive. Without this, publication is not warranted. More careful statistical analysis is needed in many of the experiments. Furthermore, no comparison between RG7716 and the combination treatment of anti-ANG-2 and anti-VEGF antibodies is shown.

Specific comments:

1. An obvious question is, whether the RG7716 CrossMab is more effective than an equivalent total amount combination of anti-VEGF and anti-ANG-2 monospecific antibodies, but this was not assessed in the manuscript. In order to evaluate the benefit of the CrossMab strategy, the authors should use, in similar experiments as presented in Figures 1, 2, 3, 7 and 8, the combination of anti-VEGF and anti-ANG-2 antibodies and compare this with the RG7716 treatment and with the anti-VEGF and anti-ANG-2 monotherapies.
2. A major problem in evaluating the data is the lack of statistical information. The number of independent experiments (n) and p values between all treatments (significant, non-significant) should be indicated for each experiment. In addition, the authors should indicate in each figure if the error bar represents SEM, as stated in the materials section.
3. The antibody dosing varies between the experiments, and within a single experiment between the treatments. A more systematic analysis should be provided, using a range of antibody concentrations for anti-VEGF, anti-ANG-2, the combination of anti-VEGF and anti-ANG-2, and for RG7716, e.g. in Figure 1G-H and 2B.
4. One additional problem is that the genes and mechanisms that lead to the neovascular phenotype in the JR5558 mouse model are not known. Thus it is not clear how well it corresponds to choroidal neovascular disease in humans.
5. Figure 1. Why were the antibodies used at 5 mg/kg in the early intervention and as 3 mg/kg in the late intervention models? Perhaps increasing the dose would improve the results in the late intervention model.
6. Figure 1E-F. Is the difference between RG7716 and anti-VEGF-A, or anti-VEGF-A and control treatment significant? - RG7716 should be compared with anti-ANG-2, as well as with the combination of anti-VEGF and anti-ANG-2 antibodies, in order to determine whether RG7716 is better than anti-ANG-2 antibody or the combination of anti-VEGF and anti-ANG-2 antibodies.
7. Figure 1G. Did any of the treatments result in statistically significant differences? If not, the text on p. 5 should be modified: "All regimens at 3 mg/kg showed a slight reduction in lesion number compared to isotype or phosphate buffered saline (PBS) control (Fig. 1G)." It should be stated that there was only a trend towards reduced lesion number, and that this was not statistically significant.
8. Figure 1H. Please indicate all p values and comparisons between treatments. Is there a statistically significant difference between RG7716 and anti-ANG-2, RG7716 and anti-VEGF, or RG7716 and control treatments, i.e. is RG7716 better than monospecific antibodies?

9. Figure 2B. Please indicate all statistically significant comparisons. Is there a statistically significant difference between anti-VEGF (5mg/kg), anti-ANG-2 (5 mg/kg), RG7716 (1 or 5 mg/kg) or Ig control? Why was RG7716 used at 10 mg/kg, which gives a significant result, whereas anti-VEGF and anti-ANG-2 were used as 5 mg/kg? The combination of anti-VEGF and anti-ANG-2 should be studied.

10. Figure 3. Anti-ANG-2 treatment should be presented in 3A. All statistical comparisons that are significant should be indicated in 3B. The combination of anti-VEGF and anti-ANG-2 should be studied.

11. Figure 6A. Error bars should be included.

12. Figure 7. The combination of anti-VEGF and anti-ANG-2 would be essential to confirm the benefit of RG7716 over combination of monospecific antibodies.

13. Figure 8. All statistically significant comparisons should be indicated in 8A-E.

Minor comments:

14. Correct sentences on page 3: the last sentence of the 2nd paragraph ("manifesting edema and/or neovascularization neovascular eye diseases") as well as the 2nd sentence of the 3rd paragraph ("Both ligands interact with the Tie-2 transmembrane receptor tyrosine kinase Tie-2").

15. Page 3, 3rd paragraph, row 7: "Endothelial cells remain present, which makes the phenotype distinct from both VEGFR1- and VEGFR2-deficient mice (Sato et al, 1995)." This needs to be corrected, as endothelial cells are present and actually even more abundant in VEGFR1-deficient mice.

16. Page 3, 3rd paragraph, row 20: "In contrast, mice engineered to inducibly overexpress ANG-2 in the endothelium show vascular leakage and loss of capillary-associated pericytes, which progressed to sepsis-like phenotype (Ziegler et al, 2013)." - Perhaps "...which progressed to sepsis-like hemodynamic alterations." would be more appropriate description here. In addition, it should be emphasized that the transgenic mice used in this study expressed human ANG-2. Thus the phenotype may include a contribution by a potential inflammatory response against the human protein expressed in adult mice.

We thank the reviewers for their helpful comments, it helped us greatly to improve our manuscript. We submit now a second version which was adapted according to these comments. We added new experiments, appropriate controls to existing ones and have rewritten/reorganised the text. We feel this has dramatically improved our manuscript and addresses the concerns of the reviewer, in detail:	
Referee #1 (Remarks):	
Regula et al have examined the effect of the bispecific VEGF and Ang2 targeting CrossMAb for ocular diseases in rodent and nonhuman primate models. The authors intended to demonstrate that targeting VEGFA and ANG2 at the same time is more effective than monotherapy. Overall, the study is a solid piece of work. However, its impact is somewhat limited because of a less than perfect experimental design, a limited scope and novelty of the results and the study's overall limited mechanistic impact.	We have now introduced single anti-VEGF-A and single ANG-2 treatment arms for the figures in the paper. We added a mechanistic study using a cellular barrier function study, a mechanistic inflammation model and discuss the cooperative aspect of VEGF-A and ANG2 biology in the paper in detail.
First, it is not clear to the reviewer if the authors have fully considered the concepts of bioequivalence. The crossmab appears to be of similar MW and behavior as the parent IgG. Yet, has this been fully validated in terms of antibody efficacy and pharmacokinetics? Apparently, the authors use equal molar concentrations in some experiments and equal weight in others. Likewise, the employed concentrations appear to vary quite significantly from experiment to experiment (see below).	Yes we have taken the concept of bioequivalence very serious. The Crossmab is a human IgG1 which binds two antigens, otherwise it is identical in structure to a human wild type IgG1. The Crossmab concept is established und referenced in our paper. We use equal molar weight in the paper now and point out the fact of equal molar concentrations or even equal molar number of antigen binding sites.
Second, it should be standard to compare the bispecific antibody with each single targeting antibody alone. However, the proper controls are missing in many experiments, most importantly the anti-ANG2 alone treatment group (see below).	Anti-ANG-2 alone is added now where it was missing. Representative pictures for all treatment groups are shown now all the times.
Third, the authors only mention statistical analysis in the "laser-induced CNV in Cynomolgus monkeys" experiment, leaving out the statistical analysis in other experiments. And some differences, which may be statistically significant, are not marked as such (see below).	Detailed statistical analysis is added now to each figure.
Beyond these technical limitations, the rationale of targeting VEGFA and ANG2 simultaneously is not solidly supported by the results. Fig 1G shows that anti-VEGFA/ANG2 did not have a superior effect in reducing lesion numbers compared to anti-VEGFA or anti-ANG2, respectively. However, anti-VEGFA/ANG2 and ANG2 were better at reducing lesion size as shown in Fig 1H. This argues that it may possibly be a better strategy to target VEGFA and ANG2 in a sequential manner.	We believe our data solidly show that simultaneous inhibition is superior to anti-VEGF alone. We used the spontaneous CNV models in great detail (from key angiogenic read outs to consequences of aberrant angiogenesis) and confirmed in two more models on key angiogenic read outs (OIR and laser CNV in NHP). Fig1G (now 2L,M) shows the experiment when the antibody is given in later intervention. Lesions have established at this stage. Therefore the antibody has little efficacy in reducing the amount of established lesions. Instead in the early intervention we

	demonstrate that the antibody is capable to prevent lesion establishment. Here the bispecific antibody works much better than monotherapy. In the late intervention we show a weak reduction of established lesion but the effect on lesion leakiness is strong, demonstrating the potential of targeting these pathways for pharmacological interference. We added OIR as second rodent model to demonstrate the superiority of the dual inhibition.
Last but not least, the manuscript is not written in a well-organized and concise manner. The results should be re-organized into more specific sections with clearer subtitles. The discussion is too redundant. The authors should discuss interpretations of the results, not summarizing related findings. In short, better evidence will need to be provided, proper controls need to be included and the manuscript should be re-written in order in order to make it a solid contribution.	We agree with the reviewer, our first version had a lot of room to improve. We believe we are now submitting a very much improved version which starts with the human data now and the mechanistic model of barrier breakdown. An acute inflammation challenge model is added to strengthen the anti-inflammatory activity of the combination. The discussion is much more focussed on our results and controls are included for all experiments as requested.
Beyond these general comments, the authors should consider the following specific suggestions:	
1. The authors have examined different aspects of anti-VEGF-A/ANG-2 in the spontaneous CNV mouse model, which demonstrated promising efficacy. However, the experiments do not appear to be designed consistently.	We added the anti-ANG-2 treatment and extended the dose response to the early intervention to have a consistent design in the JR5558 mice experiments.
1a) The dosages used in different experiments appear to be somewhat inconsistent. The authors administrated 5 mg/kg of antibodies in early intervention model and 3 mg/kg in the late-intervention model without giving reasons for the change of the dosage. To check the edema and the inflammation in the late interference model, the author then went for a range of concentrations of anti-VEGF-A/ANG-2, again without considering the previous dosage of 3 mg/kg.	We started with the early preventative experiments and higher doses. As efficacy was demonstrated we reduced the antibody concentrations and progressed to the intervention model, when lesions are already established with a lower dose which was efficacious in the preventative setting.
1b) Proper controls are missing in some of the experiments. It should be consistent throughout the manuscript to compare the bispecific antibody with each of the single targeting reagents. However, the anti-ANG-2 treatment was left out in the early intervention model (Fig 1E and 1F) and macrophage infiltration staining (Fig 3A, 3C-E). The author mentioned in the early intervention model vehicle control, which could not be found in the figures and is also not included in other experiments.	All single antibody treatments are added now. We used lesion number and area as initial read out to confirm that anti-IgG control was not different to vehicle and progressed to different read outs with anti-IgG control only.
1c) Some representative images are missing: in Fig 2A, anti-ANG-2 and anti-VEGF-A/ANG-2 at mid and high dosage; in Fig 3C-H, anti-	We added the high, mid and low dose as demonstration in Fig. 2. Figure three is spilt into two figures now to account for the

ANG-2 and anti-VEGF-A/ANG-2 at low and high dosage; in Fig 3I, anti-VEGF-A and anti-ANG-2.	request. All treatments are shown with demonstrative pictures.
1d) The authors did not state clearly in the figure legends what the error bars refer to (SD or SEM). For some apparently obvious differences, there is no significance level marked. Fig 1H: there was a significant downregulation of anti-VEGF-A treatment but not anti-VEGF-A/ANG-2, which was more effective and with smaller error bar. Fig 2A demonstrated that low dosage of anti-VEGF-A/ANG-2 already had an obvious effect, whereas in Fig 2B there is no significance detected in the low and middle dosage of anti-VEGF-A/ANG-2 treatment.	All figures have clearly stated stats in the legend. Figure 1d is correctly recognized by the reviewer. It was wrongly labelled and is corrected now. Apologies
2. The rationale of targeting VEGF-A and ANG-2 simultaneously is not substantially supported by the results in the late-intervention spontaneous CNV mouse model. The authors show that anti-VEGF-A/ANG-2 was not better than either anti-VEGF-A or anti-ANG-2 in reducing lesion numbers and not better than anti-ANG-2 in reducing lesion area. On thus wonders, if it would not be better to target VEGF-A and ANG-2 in a sequential manner?	Sequential inhibition of VEGF-A and ANG-2 in clinical practice is difficult to achieve. We therefore investigated if dual inhibition is more efficacious than anti-VEGF-A alone.
To demonstrate the superiority of the bispecific antibody and clearly lay out a good rationale, the authors definitely need to check the temporal expression of local VEGF-A and ANG-2 during the development of the disease and compare all the different targeting strategies in this aspect.	We added ANG-2 mRNA expression in the JR5558 mice model. For the VEGF expression we like to point to Nagai et al, IVOS 2014 May 20;55(6):3709-19
3. The authors generated the RG7716 with ANG-2i-LC10 binding domain. However, it was the ANG-2i-LC06 binding domain that was characterized in the cited publication (Schaefer et al, 2011). Why did the authors use the ANG-2i-LC10 instead of ANG-2i-LC06 here?	LC06 is a derivate of LC10. LC10 has not detectable binding to ANG-1 which is crucial for the mode of action as restoring a high ANG-1/ANG-2 ratio is key for the mechanism.
4. The authors addressed the question how the antibodies affect inflammation around the lesions by examining IBA-1-positive macrophages. IBA-1 is not a macrophage-specific marker. Therefore, the authors are asked to use another marker to demonstrate the results, for example CD68 or F4/80. In addition, as the authors have discussed, Tie2-positive macrophages can promote neoangiogenesis and more recently Ang2 has also been shown to recruit CCR2 positive macrophages via CCL2 and affect inflammatory pathways like NF-kb and Stat3 signaling. Hence, it would be interesting to further demonstrate which subpopulations of macrophages were affected by antibody	We have added a direct proinflammatory challenge model in the supplementary figures which shows the anti-inflammatory activity of the combination to support the concept of anti-inflammatory activity of the combination. Representative figures are shown for each treatment now and isolectin-B4 was used as marker. Indeed an additional marker can be employed. However the typical endothelial staining pattern of isolectin-B4 demonstrates the concept of co-localisation of IBA-4 and isolectin which was the objective of this figure.

treatment and provide more mechanistic insights. Isolectin is also not an endothelial specific marker. It can also be taken up by macrophages, making it unsuitable for such an analysis. Hence, another marker such as CD31 should likely be used instead. In addition, the authors claim in the text to stain endothelial cells with isolectin-B4. However, they wrote endomucin staining in the figure legend. Which marker did the authors actually use? The authors should also provide representative images of all treatment groups.	
5. In the pharmacokinetic study of RG7716 in Cynomolgus monkeys, only concentrations for RG7716 were measured in vitreal humor, without proper controls. The same experiment including anti-VEGF-A/ANG-2-FcγR as comparison is therefore required.	The main purpose of the PK study was to enable detailed dose predictions for RG7716 in a phase I study in humans and confirm the accelerated systemic clearance by comparing to published data for wild type IgG molecules. We compare exposure of a wild type IgG1 and a FAB to RG7716 (with FcRn binding modification) in the CNV efficacy study (Appendix figure S3). We limited the use of NHP to critical experiments only.
6. The authors have done substantial work to examine the bispecific antibody which shows promising effects. Could the author provide some more mechanism-based insights to explain how the antibody achieves such an effect?	We have added the mechanistic inflammation challenge model which demonstrated direct anti-inflammatory activity of the combination and explains the anti-inflammatory activity detected in the CNV models (e.g. macrophage numbers in JR5558 model and aqueous cytokines in laser CNV). Furthermore we add a mechanistic cellular model of VEGF-A induced barrier breakdown demonstrating the beneficial effect of blocking ANG-2 but also direct protective effect of ANG-1. Rebalancing the ratio of ANG-1 to ANG-2 towards high ANG-1 levels is now discussed in detail in the manuscript.
Referee #2 (Remarks):	
In their manuscript the authors present data on CrossMAb RG7716, a bispecific domain exchanged monoclonal antibody capable of binding, neutralizing, and depleting both VEGF-A and angiopoietin-2 (ANG-2). The authors use the spontaneous choroidal neovascularization (CNV) model in JR5558 mice and a nonhuman primate model of laser-induced CNV. The authors show that the simultaneous inhibition of VEGF-A and ANG-2 using RG7716 is effective in reducing vessel leakiness, CNV lesion number and in improving visual functionality - in some experiments more effectively than anti-VEGF or anti-Ang2 monotherapy. The authors also demonstrate modifications of the FcRn and FcγR binding sites of RG7716, which prevent Fc-mediated effector functions of RG7716,	

resulting in increased systemic, but not ocular, clearance.	
The manuscript deals with an important and interesting topic. There is an unmet need for better treatment options in CNV-associated diseases, such as age-related macular degeneration (AMD), since not all patients respond to current VEGF blocking therapies. In addition, previously published work has demonstrated that ANG-2 is an important contributor in CNV. However, based on the work by Regula et al., the real benefit of RG7716 in comparison to anti-VEGF or anti-Ang2 monotherapies remains elusive. Without this, publication is not warranted. More careful statistical analysis is needed in many of the experiments. Furthermore, no comparison between RG7716 and the combination treatment of anti-ANG-2 and anti-VEGF antibodies is shown.	There is an important translational aspect to our work as we try to identify target combinations which may improve VEGF-A mono therapy. The focus of our manuscript is to demonstrate that combined inhibition of VEGF/ANG-2 is more efficacious than VEGF-A inhibition alone, as anti-VEGF-A monotherapy is the standard of care in AMD. We have now added another two models showing improved efficacy of the combined inhibition. One is a classic angiogenesis model (OIR) and shows that dual inhibition is better than anti-VEGF-A alone confirming what we have seen in JR5558 mice. We also add a mechanistic inflammatory model which confirms the anti-inflammatory activity of the combination seen in the CNV models. Here we demonstrate direct anti-inflammatory activity of the combination and demonstrate equal efficacy of the CrossMab to single antibody combination. This part is discussed in the manuscript in detail now. Finally we add a cellular experiment which demonstrates a key mechanism of dual inhibition namely the rebalancing of the ANG-2/ANG-1 ratio with increased ANG-1 restoring barrier function and counteracting neoangiogenic signals. Detailed statistical analysis is added now for each experiment.
1. An obvious question is, whether the RG7716 CrossMAB is more effective than an equivalent total amount combination of anti-VEGF and anti-ANG-2 monospecific antibodies, but this was not assessed in the manuscript. In order to evaluate the benefit of the CrossMAB strategy, the authors should use, in similar experiments as presented in Figures 1, 2, 3, 7 and 8, the combination of anti-VEGF and anti-ANG-2 antibodies and compare this with the RG7716 treatment and with the anti-VEGF and anti-ANG-2 monotherapies.	We added supplementary figure 2 which tests the anti-inflammatory action of the VA2 combination. While the single reagents are only weakly active we demonstrate that combined inhibition is better than the single reagents. In this experiment we also directly compare the CrossMAB to the combined inhibition using monoclonal antibodies. While both groups are more efficacious than single treatments, the combination is equally efficacious no matter if using CrossMAB or two monoclonal antibodies. Again we like to stress we are not arguing that a CrossMAB is more efficacious as two monoclonal antibodies, we argue dual VEGF/ANG-2 inhibition is more efficacious than VEGF-A alone.
2. A major problem in evaluating the data is the lack of statistical information. The number of independent experiments (n) and p values between all treatments (significant, non-significant) should be indicated for each experiment. In addition, the authors should indicate in each figure if the error bar represents SEM, as stated in the materials	Done and apologies for not having it done in version one already

section.	
3. The antibody dosing varies between the experiments, and within a single experiment between the treatments. A more systematic analysis should be provided, using a range of antibody concentrations for anti-VEGF, anti-ANG-2, the combination of anti-VEGF and anti-ANG-2, and for RG7716, e.g. in Figure 1G-H and 2B.	We have added a dose response of dual inhibition in Fig. 1 (now Fig 2) showing the superior efficacy of dual inhibition compared to a higher dose of single treatments. Likewise the anti-ANG2 alone is added as comparator. The early intervention experiments encouraged us to take a low dose of bispecific antibody into the late intervention study when disease is already much more established and interference at this stage reflects more the clinical situation.
4. One additional problem is that the genes and mechanisms that lead to the neovascular phenotype in the JR5558 mouse model are not known. Thus it is not clear how well it corresponds to choroidal neovascular disease in humans.	We agree to this statement. We are currently working toward understanding the genes and mechanism behind the JR5558 mouse models. We demonstrate efficacy in the laser induced model in the NHP in addition to the JR5558 mice. This is a more widely established model of CNV, however we argue the translatability of that model has also its limitations. We therefore add a third model (OIR) in which we show better efficacy of dual inhibition compared to anti-VEGF-A only (Supplemental figure 1). We therefore argue that the overall evidence is supportive of testing this concept in the clinic.
5. Figure 1. Why were the antibodies used at 5 mg/kg in the early intervention and as 3 mg/kg in the late intervention models? Perhaps increasing the dose would improve the results in the late intervention model.	We demonstrate now 3 mg/kg is also efficacious in the early intervention model. As efficacy was demonstrated as low as 3 mg/kg in the early intervention, we took the lowest dose into the more challenging model of later intervention and could still demonstrate good efficacy of the combination.
6. Figure 1E-F. Is the difference between RG7716 and anti-VEGF-A, or anti-VEGF-A and control treatment significant? - RG7716 should be compared with anti-ANG-2, as well as with the combination of anti-VEGF and anti-ANG-2 antibodies, in order to determine whether RG7716 is better than anti-ANG-2 antibody or the combination of anti-VEGF and anti-ANG-2 antibodies.	The single ANG-2 treatment is tested in Figure 1 now (see Fig 2 now). We wanted to preclinically test the concept of dual inhibition of VEGF-A & ANG2 being better than anti VEGF-A alone. To our mind this can be tested equally well with either a CrossMAb or single antibody combinations preclinical.
7. Figure 1G. Did any of the treatments result in statistically significant differences? If not, the text on p. 5 should be modified: "All regimens at 3 mg/kg showed a slight reduction in lesion number compared to isotype or phosphate buffered saline (PBS) control (Fig. 1G). " It should be stated that there was only a trend towards reduced lesion number, and that this was not statistically significant.	The concept of equivalent inhibition using a CrossMAb versus single reagents was established in earlier work (Schaefer et al, 2011). It was our objective to demonstrate that dual inhibition of VEGF-A and ANG-2 is more efficacious than VEGF-A alone. Fig 1G: Now Fig 2L yes the reduction of the antibody is smaller, as lesion have established already when interfering late with the antibody. However, changes are significant. The effect of dual inhibition is stronger when it comes to lesion area. This we argue is a key to the VEGF-A/ANG2 blockade which is strong on reducing permeability and edema.

8. Figure 1H. Please indicate all p values and comparisons between treatments. Is there a statistically significant difference between RG7716 and anti-ANG-2, RG7716 and anti-VEGF, or RG7716 and control treatments, i.e. is RG7716 better than monospecific antibodies?	Done, now Fig 2 J-K The high dose of the anti-VEGF-A/ANG-2 CrossMAB achieves significance against IgG control for all measures. No single reagent achieves this. In the early intervention the high dose of the anti-VEGF-A/ANG-2 CrossMAB is significant compared to anti-VEGF-A and anti-ANG-2 alone.
9. Figure 2B. Please indicate all statistically significant comparisons. Is there a statistically significant difference between anti-VEGF (5mg/kg), anti-ANG-2 (5 mg/kg), RG7716 (1 or 5 mg/kg) or Ig control? Why was RG7716 used at 10 mg/kg, which gives a significant result, whereas anti-VEGF and anti-ANG-2 were used as 5 mg/kg? The combination of anti-VEGF and anti-ANG-2 should be studied.	Done, now figure 3b. We like to draw the attention to the fact that a normal IgG has two binding sites for a ligand compared to a crossmab, Therefore 10 mg/kg of CrossMAB delivers the same number of ligand binding sites as 5 mg/kg of a normal IgG.
10. Figure 3. Anti-ANG-2 treatment should be presented in 3A. All statistical comparisons that are significant should be indicated in 3B. The combination of anti-VEGF and anti-ANG-2 should be studied.	Now Fig 4: anti-ANG-2 treatment is added, figures and stats added. In supplementary figure S2 we compared the CrossMAB to single treatments and did not see any difference in efficacy. This confirms the concept of equal efficacy in blockade using a CrossMAB versus single reagents.
11. Figure 6A. Error bars should be included.	Done
12. Figure 7. The combination of anti-VEGF and anti-ANG-2 would be essential to confirm the benefit of RG7716 over combination of monospecific antibodies.	We demonstrate that dual inhibition of VEGF and ANG-2 is more beneficial than VEGF-A alone but it is not our aim to demonstrate that a CrossMAB targeting VEGF and ANG-2 is better than its individual components.
13. Figure 8. All statistically significant comparisons should be indicated in 8A-E.	Done now Fig9
14. Correct sentences on page 3: the last sentence of the 2nd paragraph ("manifesting edema and/or neovascularization neovascular eye diseases") as well as the 2nd sentence of the 3rd paragraph ("Both ligands interact with the Tie-2 transmembrane receptor tyrosine kinase Tie-2").	Done
15. Page 3, 3rd paragraph, row 7: "Endothelial cells remain present, which makes the phenotype distinct from both VEGFR1- and VEGFR2-deficient mice (Sato et al, 1995)." This needs to be corrected, as endothelial cells are present and actually even more abundant in VEGFR1-deficient mice.	Done
16. Page 3, 3rd paragraph, row 20: "In contrast, mice engineered to inducibly overexpress ANG-2 in the endothelium show vascular leakage and loss of capillary-associated pericytes, which progressed to sepsis-like phenotype (Ziegler et al, 2013)." - Perhaps "...which progressed to sepsis-like	We agree to this suggestion of sepsis-like hemodynamic alteration and changed accordingly, the human ANG-2 expression versus rodent should not be an inflammatory issue, as new born mice will have undergone germ line selection.

hemodynamic alterations." would be more appropriate description here. In addition, it should be emphasized that the transgenic mice used in this study expressed human ANG-2. Thus the phenotype may include a contribution by a potential inflammatory response against the human protein expressed in adult mice.	
---	--

Schaefer W, Regula JT, Bahner M, Schanzer J, Croasdale R, Durr H, Gassner C, Georges G, Kettenberger H, Imhof-Jung S et al (2011) Immunoglobulin domain crossover as a generic approach for the production of bispecific IgG antibodies. Proceedings of the National Academy of Sciences of the United States of America 108: 11187-11192

Thank you for the submission of your revised manuscript to EMBO Molecular Medicine. We have now heard back from the reviewers whom we asked to evaluate your manuscript.

You will see that both reviewers remain reserved and point to a number of important pending issues that would require adequate action. The concerns touch upon data quality, lack of clarity and experimental issues including internal inconsistencies and discrepancies.

I am especially concerned about the issue mentioned by both reviewers on the reliability of experimentation based on the very long EC culture time in the starvation medium and in the culture inserts for TER measurements. The remaining concerns on statistical treatment are also worrying as the general issue is close to our hearts here at EMBO Press.

Although we would normally not allow a second significant revision, following our internal discussions I am prepared in this case, to give you the opportunity to improve your manuscript by carefully responding to each point and providing additional experimental evidence where necessary as mentioned above. It is mostly likely that I will have to refer back to the reviewers on your next, final version.

As you know, EMBO Molecular Medicine has a "scooping protection" policy, whereby similar findings that are published by others during review or revision are not a criterion for rejection. However, I do ask you to get in touch with us after three months if you have not completed your revision, to update us on the status. Please also contact us as soon as possible if similar work is published elsewhere.

I look forward to seeing a revised form of your manuscript as soon as possible.

***** Reviewer's comments *****

Referee #1 (Remarks):

This reviewer fully acknowledges that the authors have taken great effort to substantially improve the manuscript in line with the reviewers' comments. The revised figures now include all conditions and statistics. This is all much more convincing and meaningful. The additional mechanistic studies provide more insight. Lastly, the discussion has been substantially improved. However, there are still a number of open questions that need to be addressed.

The concentrations of antibodies are not consistent throughout the study. The authors argue that they lowered the antibody concentration in the late intervention model because they observed good efficacy of the antibody in the early intervention scheme. However, this line of arguments is not fully convincing: A lower antibody concentration was used in the late intervention trial, where the symptoms are more severe. In turn, a high dose was used in the newly added hyperoxia model (Fig. S1), in which the authors employed a single treatment that was not used before and the authors did not consider the bioequivalence of the different reagents.

The reviewer is somewhat concerned with the quality of the data shown in Fig. 10. The data in the new manuscript are presented as log scale, which makes it difficult to compare them with the original manuscript in which a linear scale was used. Yet, it is the reviewer's impression that the values and even trends are not consistent between the two manuscript, especially anti-ANG-2 and anti-VEGF treated groups. How would the authors explain such the discrepancies? In additions, it is appreciated that the authors now include the statistical analysis in the figure. Yet, samples collected at different days from differently treated animals should likely not be compared with two variables. The authors also mentioned that aqueous humor was collected at baseline in "Materials and Methods", which is not included in Fig. 10.

Following are few more specific comments related to the individual figures:

Fig.1: Fig.1E shows transendothelial electrical resistance measurements of cells cultured for more than 4 days prior to measurement. How did the cells survive for 4 days in the culture inserts? The figure also shows a sharp decrease in the resistance of the control group that likely reflects cell death rather than a proper control for the experiment. How do the authors justify the design and adequacy of the experiment lacking a viable control? Which endothelial cells were used for the experiment? The asterisks representing the statistical analysis in Fig. 1A and B should likely be enlarged.

Fig.2: Fig.2A shows an increase in ANG-2 during the later stages of disease. However, the early intervention shows a better effect. How do the authors explain this discrepancy? This point should be discussed in further detail in the manuscript. Anti-ANG-2 treatment during late intervention results in a similar effect as the combined inhibition. Please address that in the discussion.

The authors mention that they observed no difference between vehicle and IgG control.

Consequently, the vehicle control is not included in later studies. This should also be addressed in the manuscript. In addition, statistical analysis of vehicle control compared to the treatment groups is missing in Fig.2M.

The authors should clarify in Fig. 2J-M more specifically if area or number of lesions was measured.

Fig.3: The picture, especially of the anti-ANG-2 treatment, shown in Fig.3B is not representative of the quantification shown in Fig.3A.

Fig.4: The fact that IBA-4 is not a macrophage-specific marker and the suggestion of using another marker, if one wants to stain changes in macrophage number, was not considered even though this was already mentioned in the review of the original manuscript.

Fig.5: Please label panel "I" with the corresponding letter.

Fig.8: The single anti-ANG-2 treatment is missing in panel A.

Fig.9: Representative images of all treatments should be included.

Fig.S1: Again, the single anti-ANG-2 treatment is missing in panels B and C.

Fig.S2: Panel C excludes the quantification of the challenge only and the CrossMab antibody group that are included in panel B.

As a minor point, it is suggested to include subtitles if a section consists of more than one paragraph. Please consider removing the subtitle on page 5.

Referee #2 (Remarks):

The authors have answered the majority of the questions raised by this reviewer, and the revised manuscript by Regula et al. has significantly improved. However, few issues deserve consideration.

Author response to comment 1. The authors should discuss, in the discussion, their conclusion that CrossMab is more efficacious than anti-VEGF alone, but not more efficacious than anti-Ang2+anti-VEGF.

Author response to comment 2. The statistical presentation of the results has improved. However, the significance is not provided for comparisons between all treatment groups in all figures, potentially because these were not significant. For clarity, please state in each figure legend if all significant comparisons are indicated, and all non-significant comparisons are omitted, or alternatively, indicate all comparisons (non-significant with NS).

Author response to comment #3. The authors state that Fig. 2 (J, K) demonstrates superior efficacy of dual inhibition compared to a single treatments and that anti-Ang2 is included. However, the statistical comparison between CrossMab (high dose) and anti-Ang2 is missing in 2J. Also, it would be helpful for the reader to distinguish between J and K as well as L and M by modifying the Y-axis legend as "Lesion number" and "Lesion area", accordingly.

Author response to comment #16. As embryonic Ang2 expression is lethal, Ang2 expression is induced postnatally in the transgenic model.

Figure 10. The statistical comparisons require clarification. E.g. in 10B, why is RG7716 D16 compared to IgG control D30 or to anti-Ang2 D16? In 10C, why is Ang2 D16 compared with anti-VEGF D30 and RG7716 D30, and not D16 values? 10D: the comparison between anti-Ang2 D16 and anti-Ang2 D30 is indicated twice.

p. 4 Results, second last row. "... demonstrating vessel normalization function of Ang1." - As cultured cells were used, this phrase should be corrected, e.g. "...improved endothelial monolayer integrity".

p. 11 Discussion, 3rd paragraph. As Vasculotide was recently reported not to bind to Tie2, the mechanisms of action of Vasculotide should be interpreted with caution (Wu et al., Embo Mol Med, 2015).

p. 15. Materials and Methods, 2nd paragraph. Why did the authors culture endothelial cells for 2 more days in the starvation media before analysis of monolayer permeability? This seems to be a long period in the absence of growth factors and in low serum concentration.

2nd Revision - authors' response

20 July 2016

We thank the reviewers for their comments to our updated manuscript. Again their comments helped us to improve the manuscript further. In detail we like to comment on the points raised.

Reviewer 1:

This reviewer fully acknowledges that the authors have taken great effort to substantially improve the manuscript in line with the reviewers' comments. The revised figures now include all conditions and statistics. This is all much more convincing and meaningful. The additional mechanistic studies provide more insight. Lastly, the discussion has been substantially improved. However, there are still a number of open questions that need to be addressed.

The concentrations of antibodies are not consistent throughout the study. The authors argue that they lowered the antibody concentration in the late intervention model because they observed good efficacy of the antibody in the early intervention scheme. However, this line of arguments is not fully convincing: A lower antibody concentration was used in the late intervention trial, where the symptoms are more severe. In turn, a high dose was used in the newly added hyperoxia model (Fig. S1), in which the authors employed a single treatment that was not used before and the authors did not consider the bioequivalence of the different reagents.

The reviewer is somewhat concerned with the quality of the data shown in Fig. 10. The

We compare the bispecific antibody at three concentrations to anti-VEGF-A and anti-ANG-2 in the JR5558 mouse model. The 10 mg/kg concentration offers the same molar concentration of anti-VEGF-A binding sites as the anti-VEGF-A at 5 mg/kg as it is an IgG1. All concentrations of anti-VEGF/ANG-2 used are therefore the same or lower than anti-VEGF-A. The comparison of three concentration bispecific in dose response consistently favors the bispecific to being more efficacious compared to anti-VEGF-A alone.

In response to the reviewer that we only employed one rodent model we added OIR. We demonstrate that one concentration of bispecific is superior to anti-VEGF-A, as we did not had previous experience with our antibodies in this model we chose a higher concentration (10mg/kg) and indeed this one confirmed the better efficacy of the combination compared to anti-VEGF-A only. While the anti-VEGF-A was active the combination was significantly better. Using 10 mg/kg, we even used half the molar concentration of anti-VEGF-A binding sites in the bispecific antibody. The experiment is therefore not biased toward the bispecific approach.

data in the new manuscript are presented as log scale, which makes it difficult to compare them with the original manuscript in which a linear scale was used. Yet, it is the reviewer's impression that the values and even trends are not consistent between the two manuscript, especially anti-ANG-2 and anti-VEGF treated groups. How would the authors explain such the discrepancies? In additions, it is appreciated that the authors now include the statistical analysis in the figure. Yet, samples collected at different days from differently treated animals should likely not be compared with two variables. The authors also mentioned that aqueous humor was collected at baseline in "Materials and Methods", which is not included in Fig. 10.	We agree with the reviewer that the linear scale is better to compare and revert to this presentation. For the statistics analysis we have to use logarithmically transformed data as the cytokine data are not normally distributed. If all significant changes are shown indeed day16 and day30 analysis dominate which are not meaningful. We therefore performed two analyses separately for each time point. We highlight the fact that the stats is done on transformed data in the text. The baseline data were from before the experiment started and showed that no IL-6 and IL-8 was detectable in the aqueous. The data don't add value to the experiment at this stage and we removed the comment in the material and methods.
Fig.1: Fig.1E shows transendothelial electrical resistance measurements of cells cultured for more than 4 days prior to measurement. How did the cells survive for 4 days in the culture inserts? The figure also shows a sharp decrease in the resistance of the control group that likely reflects cell death rather than a proper control for the experiment. How do the authors justify the design and adequacy of the experiment lacking a viable control? Which endothelial cells were used for the experiment? The asterisks representing the statistical analysis in Fig.1A and B should likely be enlarged.	The cells survive the conditions very well, the cells still have 0.5% FCS in a medium designed for low FCS culture and all growth factor supplements are still present except VEGF-A which is a critical growth factor for the growth of the cells. However to perform our studies VEGF-A needs to be taken out VEGF-A of the system. As a strong heparin/extracellular matrix molecule we give the cells enough time to clear VEGF-A completely out the system before restimulation. We confirmed the cells are alive and we can exclude apoptosis as a mechanism. The control group keeps a high barrier which is an energy dependent process. Only the VEGF-A addition induced barrier breakdown which is reduced with ANG-1 and anti-ANG-2. We added a supplementary figure (Fig. S1) which shows that addition of RG7716 at the end of the experiment fully restores VEGF-A induced barrier breakdown which shows clear VEGF dependence and can exclude apoptosis as mechanism. We add a figure for the reviewer below showing that cells are alive for 7 days under our culture conditions. We measured cell viability with alamar blue of cells 3, 5 and 7 days on filters under assay conditions as used to measure TEER and compared EGM-2 growth medium with 2% FBS (Lonza) to EGM-2 with 0.5% FBS with supplements except VEGF-A and EBM-2 containing 0.5%FBS only. Only EBM-2 cell culture conditions demonstrated lower cell viability. However we believe Fig S1 is more relevant for a broader audience and added it as a supplemental figure. We change the term hunger medium to assay medium to avoid the impression the medium is not allowing cell survival.

The asterisks are enlarged in Fig 1A and B now.

Fig.2: Fig.2A shows an increase in ANG-2 during the later stages of disease. However, the early intervention shows a better effect. How do the authors explain this discrepancy? This point should be discussed in further detail in the manuscript. Anti-ANG-2 treatment during late intervention results in a similar effect as the combined inhibition. Please address that in the discussion. The authors mention that they observed no difference between vehicle and IgG control. Consequently, the vehicle control is not included in later studies. This should also be addressed in the manuscript. In addition, statistical analysis of vehicle control compared to the treatment groups is missing in Fig.2M. The authors should clarify in Fig. 2J-M more specifically if area or number of lesions was measured.

The data are generated from RNA extracted from whole retina of which the inner retinal endothelium only makes up a small proportion. However strong and significant upregulation is seen at the late point, however increases are also seen at the earlier time point but they do not reach significance. This fits general ANG-2 biology with ANG-2 levels increasing with disease severity, as we also show stronger ANG-2 elevation comparing DR with more severe pDR.

The comparison of vehicle and IgG is highlighted now on page 6 upper chapter. All statistical comparisons are shown and the figure labelling is addressed as suggested.

We extended the discussion with this paragraph:

Late interference in the model did not reduce lesion numbers as strongly compared to early intervention, as neovascularization has taken place lesions have developed before antibody is given. Furthermore the efficacy of anti-ANG-2 alone is as strong as the bispecific anti-VEGF-A/ANG-2 treatment at reducing lesion area in later interference. The results may highlight a more prominent role for VEGF-A mediating neovascularisation and ANG-2 mediating vessel leakiness. At this stage, it is also not clear if the small reduction in leaky lesion number in the late intervention model is due to reduction of new lesions generated at the late time point or by induction of lesion regression, a mechanism of action proposed for the VEGF-A/PDGF-B dual targeting...

And

While anti-ANG-2 reduced acute vessel leakiness as measured by FA, reduction of retinal edema was stronger with by anti-

	VEGF-A/ANG-2 as measured by OCT. This observation warrants further investigation.
Fig.3: The picture, especially of the anti-ANG-2 treatment, shown in Fig.3B is not representative of the quantification shown in Fig.3A.	Changed
Fig.4: The fact that IBA-4 is not a macrophage-specific marker and the suggestion of using another marker, if one wants to stain changes in macrophage number, was not considered even though this was already mentioned in the review of the original manuscript.	We believe the reviewer is referring to isolectin-B4 which we used to stain the vasculature but used IBA-1 to stain for macrophages. In our initial assessment to identify macrophages in the JR5558 mice we demonstrated a very nice overlap of the staining of IBA-1 and F4/80, both markers are widely used to identify macrophages. We attach a figure below to this document, to our mind it does not warrant another supplementary figure. However we are happy to do so if requested. We reference three publications which also use IBA-1 as a macrophage marker compatible with anti-F4/80, anti-BM8, anti-CD68 in mouse tissue: 'Kierdorf, Katrin, et al. "Microglia emerge from erythromyeloid precursors via Pu. 1-and Irf8-dependent pathways." Nature neuroscience 16.3 (2013): 273-280.' 'Mildner, Alexander, et al. "Microglia in the adult brain arise from Ly-6ChiCCR2+ monocytes only under defined host conditions." Nature neuroscience 10.12 (2007): 1544-1553.' 'Mueller, Marcus, et al. "Macrophage response to peripheral nerve injury: the quantitative contribution of resident and hematogenous macrophages." Laboratory investigation 83.2 (2003): 175-185.'

Fig.5: Please label panel "I" with the corresponding letter.	Done
Fig.8: The single anti-ANG-2 treatment is missing in panel A.	What we describe here is a specific activity described for VEGF-A165 as a heparin binding protein and anti-VEGF-A IgGs by several authors. We aim to show with Fig 8A that this aspect of anti-VEGF-A biology does not apply to our antibody as it is effectorless and has only one anti-VEGF-A binding arm. ANG-2 has never been suggested in this context. Indeed it might be a general concept for antibodies which target heparin binding growth factors like bFGF. However this goes beyond the scope of our manuscript to our min. We therefore concentrated our investigation on the aspect that RG7716 does not show this feature described for other VEGF-A neutralizing reagents.
Fig.9: Representative images of all treatments should be included.	Representative images for all treatments have been added.
Fig.S1: Again, the single anti-ANG-2 treatment is missing in panels B and C.	Fig S1 (now FigS2) was generated to show the stronger efficacy of anti-VEGF-A/ANG-2 compared to anti VEGF-A only as this is the standard of care concept and we aimed to

	confirm that dual activity is more efficacious in a second rodent model of neovascularization. The efficacy of anti ANG-2 only in a model of OIR was demonstrated by Rennel et al 2011.
Fig.S2: Panel C excludes the quantification of the challenge only and the CrossMab antibody group that are included in panel B. As a minor point, it is suggested to include subtitles if a section consists of more than one paragraph. Please consider removing the subtitle on page 5	Fig S3 (former S2) compares now challenge and the CrossMab group. All significant changes are shown We removed the subtitle on page 5
Reviewer 2:	
The authors have answered the majority of the questions raised by this reviewer, and the revised manuscript by Regula et al. has significantly improved. However, few issues deserve consideration.	
Author response to comment 1. The authors should discuss, in the discussion, their conclusion that CrossMab is more efficacious than anti-VEGF alone, but not more efficacious than anti-Ang2+anti-VEGF.	We have adapted the discussion of bispecific versus co-formulation arguing that the key advantage of the bispecific is the single reagent that needs to be delivered when ligands are available at comparative concentrations in the tissue of interest and highlight limitation of the bispecific approach, i.e. when one ligand is in large excess (page 11 top part): A bispecific CrossMab antibody offers the advantage of having to deliver only a single molecule with a unique set of molecular properties by intravitreal injection to neutralize two targets at once, in our case VEGF-A and ANG-2. While co-formulations of antibodies or biologics using the same paratope as in a CrossMab is an alternative way to deliver therapeutic drugs. In cases where one ligand is in large excess of the other a co-formulation approach seems preferable to be able to increase the dose of the reagent neutralizing the ligand in excess.
Author response to comment 2. The statistical presentation of the results has improved. However, the significance is not provided for comparisons between all treatment groups in all figures, potentially because these were not significant. For clarity, please state in each figure legend if all significant comparisons are indicated, and all non-significant comparisons are omitted, or alternatively, indicate all comparisons (non-significant with NS).	We shown nonw only significant comparisons and omitted non-significant ones. We have introduced a sentence to each figure legends saying this. We made three exemptions were non-significant values are shown as they are approaching significance. We point to this fact in the legend as well. Fig 1D, Fig 3A, Supplement Fig 3
Author response to comment #3. The authors state that Fig. 2 (J, K) demonstrates superior efficacy of dual inhibition compared to a single treatments and that anti-Ang2 is included. However, the statistical comparison between CrossMab (high dose)	We only highlighted the difference between VEGF-A and the high dose bispecific in the last version in Fig 2J. Instead we show now all significant differences, including the single reagents compared to CrossMab high dose. The legends were modified as

and anti-Ang2 is missing in 2J. Also, it would be helpful for the reader to distinguish between J and K as well as L and M by modifying the Y-axis legend as "Lesion number" and "Lesion area", accordingly.	suggested. We also add the notion that all significant changes are shown.
Author response to comment #16. As embryonic Ang2 expression is lethal, Ang2 expression is induced postnatally in the transgenic model.	Ok I did not consider that, therefore indeed a foreign antigen response as an enhancer of response can not be excluded. On the other hand neutralizing antibody should blunt the response of human ANG-2. The authors of the paper also report classic sepsis models in which the anti-ANG-2 showed reduced severity. I would still argue the overall evidence of this paper argues for a role of ANG-2 in mediating endothelial dysfunction including barrier breakdown in sepsis.
Figure 10. The statistical comparisons require clarification. E.g. in 10B, why is RG7716 D16 compared to IgG control D30 or to anti-Ang2 D16? In 10C, why is Ang2 D16 compared with anti-VEGF D30 and RG7716 D30, and not D16 values? 10D: the comparison between anti-Ang2 D16 and anti-Ang2 D30 is indicated twice.	We agree the discussion is not meaningful. We aimed to discuss all relevant changes but by separating the analysis per day the data get more meaningful. The cytokine data are not normally distributed and therefore transformed logarithmically for the statistical analysis. If all significant changes are shown indeed day16 and day30 analysis dominates. We have therefore performed two analyses separately for each time point. We also reverted to the linear scale for better presentation and highlight the fact that the statistics is done on transformed data in the text.
p. 4 Results, second last row. "... demonstrating vessel normalization function of Ang1." - As cultured cells were used, this phrase should be corrected, e.g. "...improved endothelial monolayer integrity".	Excellent suggestion which we inserted as suggested
p. 11 Discussion, 3rd paragraph. As Vasculotide was recently reported not to bind to Tie2, the mechanisms of action of Vasculotide should be interpreted with caution (Wu et al., Embo Mol Med, 2015).	We point this out in the discussion and added the reference
p. 15. Materials and Methods, 2nd paragraph. Why did the authors culture endothelial cells for 2 more days in the starvation media before analysis of monolayer permeability? This seems to be a long period in the absence of growth factors and in low serum concentration.	The EBM culture medium used is designed for endothelial cell culture under low FCS conditions, it allows cells to be viable for the performed experiment. We adapted the medium from 2 to 0.5% FCS, kept all growth factors and supplements except VEGF-A, a key growth factor which needs to be depleted out of the system. This allows quiescence of cells and a confluent filter coverage necessary to build a TEER. When a stable TEER is achieved we start adding the barrier reducing factors. Only viable cells

	generate TEER, we can exclude apoptosis as driver of barrier breakdown. We added supplementary figure (S1) which shows the reversal of barrier breakdown induced by VEGF-A using RG7716. We give RG7716 18h after the addition of VEGF-A and obtain full reversal of the normal TEER value at the start of the experiment. This clearly shows that the cells are viable, as they restore barrier.
--	--

3rd Editorial Decision

09 August 2016

Thank you for the submission of your revised manuscript to EMBO Molecular Medicine. We have now received the enclosed reports from the referees that were asked to re-assess it. As you will see the reviewers are now globally supportive and I am pleased to inform you that we will be able to accept your manuscript pending the following final amendments:

- 1) As per our Author Guidelines, the description of all reported data that includes statistical testing must state the name of the statistical test used to generate error bars and P values, the number (n) of independent experiments underlying each data point (not replicate measures of one sample), and the actual P value for each test (not merely 'significant' or ' $P < 0.05$ ').
- 2) Please include a size bar in Fig. 5 panels B-G.
- 3) The manuscript must include a statement in the Materials and Methods identifying the institutional and/or licensing committee approving the experiments, including any relevant details (like how many animals were used, of which gender, at what age, which strains, if genetically modified, on which background, housing details, etc). We encourage authors to follow the ARRIVE guidelines for reporting studies involving animals. Please see the EQUATOR website for details: <http://www.equator-network.org/reporting-guidelines/improving-bioscience-research-reporting-the-arrive-guidelines-for-reporting-animal-research/>. I note that not all the above details are reported (including for mice).
- 4) For experiments involving human subjects the authors must identify the committee approving the experiments and include a statement that informed consent was obtained from all subjects and that the experiments conformed to the principles set out in the WMA Declaration of Helsinki [<http://www.wma.net/en/30publications/10policies/b3/>] and the NIH Belmont Report [<http://ohsr.od.nih.gov/guidelines/belmont.html>]. Any restrictions on the availability or on the use of human data or samples should be clearly specified in the manuscript. Any restrictions that may detract from the overall impact of a study or undermine its reproducibility will be taken into account in the editorial decision. Again, I note that not all the relevant information has been included in the manuscript.
- 5) We encourage the publication of source data, particularly, but not limited to electrophoretic gels and blots, with the aim of making primary data more accessible and transparent to the reader. Would you be willing to provide a PDF file per figure that contains the original, uncropped and unprocessed scans of all or at least the key gels used in the manuscript? The PDF files should be labeled with the appropriate figure/panel number, and should have molecular weight markers; further annotation may be useful but is not essential. The PDF files will be published online with the article as supplementary "Source Data" files. If you have any questions regarding this just contact me.
- 6) Every published paper now includes a 'Synopsis' to further enhance discoverability. Synopses are displayed on the journal webpage and are freely accessible to all readers. They include a short standfirst as well as 2-5 one sentence bullet points that summarise the paper. Please provide the synopsis including the short list of bullet points that summarise the key NEW findings. The bullet points should be designed to be complementary to the abstract - i.e. not repeat the same text. We

encourage inclusion of key acronyms and quantitative information. Please use the passive voice. Please attach this information in a separate file or send them by email, we will incorporate it accordingly. You are also welcome to suggest a striking image or visual abstract to illustrate your article. If you do please provide a jpeg file 550 px-wide x 400-px high.

Please submit your revised, final manuscript within two weeks. I look forward to seeing it as soon as possible.

***** Reviewer's comments *****

Referee #1 (Remarks):

Following two rounds of revision, this reviewer considers the manuscript acceptable for publication.

Referee #2 (Remarks):

No further comments

Corresponding Author Name: Guido Hartmann
Manuscript Number: TBC